# PI-CCA: PROMPT-INVARIANT CCA CERTIFICATES FOR REPLAY-FREE CONTINUAL MULTIMODAL LEARNING

**Jiayu Zhang**[1][†], **Chuangxin Zhao**[2][†], **Canran Xiao**[3][†]
**Ruibo Duan**[4], **Wenyi Mo**[5], **Haoyu Gao**[6], **Wenshuo Wang**[5][*]
[1]Peking University, [2]Institute of Automation, Chinese Academy of Sciences
[3]Shenzhen Campus of Sun Yat-sen University, [4]Seoul National University
[5]South China University of Technology, [6]Georgia Institute of Technology
`202364870251@mail.scut.edu.cn`

## ABSTRACT

When deployed on non-stationary data streams, foundation vision-language models require continual updates without access to past data. However, naive fine-tuning undermines their zero-shot recognition capabilities and prompt robustness. We seek a replay-free principle that preserves pre-trained cross-modal generalization under domain and prompt shifts. We introduce *Prompt-Invariant CCA Certificates*(PI–CCA), a geometry-first approach that summarizes image–text alignment with a compact certificate capturing the top-$k$ canonical spectrum and subspace. During adaptation, we match this summary using only mini-batch statistics and induce prompt robustness via averaging over perturbations. Across MTIL, X-TAIL, VLCL, and ConStruct-VL, PI–CCA achieves state-of-the-art performance among replay-free methods. By optimizing alignment invariants rather than proxy signals, PI–CCA provides a simple, generator-free, constant-memory path to continual adaptation with strong zero-shot retention and resilience to prompt/style shifts.

## 1 INTRODUCTION

Foundation vision–language models (VLMs) (Radford et al., 2021; Awais et al., 2025) enable zero-shot recognition and retrieval across changing domains (Patel et al., 2023; Feng et al., 2024; Yada et al., 2025; Chan et al., 2025; Zhang et al., 2025; Shi et al., 2025; Ni et al., 2025; LIU et al., 2026). In practice, they must be continually adapted to non-stationary streams without storing past data (privacy/licensing/cost), while preserving zero-shot transfer and prompt robustness—conditions that standard fine-tuning often violates (Yao et al., 2023; Li & Ke, 2025; Peng et al., 2026). This *vision–language continual learning* (**VL–CL**) setting (Zheng et al., 2023) presents two core challenges: avoiding catastrophic forgetting of cross-modal alignment (and thus zero-shot ability) and maintaining robustness to prompt/distribution shifts, typically without task IDs and under tight memory budgets (Liu et al., 2025).Overcoming these challenges could allow VLMs to be applied in a wider range of fields (Xu et al., 2025; Zhang et al., 2024; Zhan et al., 2025).

Prior VL–CL research has made notable progress via proxy constraints or architectural mechanisms: distributional/logit distillation and off-diagonal similarity alignment to stabilize representations (Zheng et al., 2023; Ni et al., 2023; Zhu et al., 2023; Cui et al., 2024; Liu et al., 2025; Gao et al., 2024), parameter-efficient or router-based adapters to separate old and new knowledge (Yu et al., 2024; Tang et al., 2024; Xu et al., 2024), and replay or stream benchmarks to mitigate data unavailability (Yan et al., 2022; Lei et al., 2023; Yao et al., 2024; Smith et al., 2023; Zhang et al., 2023; Garg et al., 2024). Yet these proxies leave a persistent structural weakness: *they regularize outcomes (similarities, logits, weights, routes) rather than directly controlling the alignment object that underlies cross-modal generalization.* As a consequence, current methods can (i) permit slow

---

[*]Corresponding author
[†]These authors contributed equally to this work

drift of the alignment geometry that drives zero-shot performance, (ii) depend on reference corpora, generators, or task metadata that are not always available, and (iii) remain brittle to prompt/style variation even when average metrics improve. This gap suggests the need for a replay-free principle that preserves alignment as an invariant, not merely as a byproduct of surrogate objectives.

We ask: *Can continual adaptation preserve cross-modal generalization by explicitly maintaining the geometry of image–text alignment, without storing past data?* Our answer is a replay-free, geometry-first framework that treats alignment as a first-class invariant and constrains its spectral and subspace structure with a compact, task-agnostic summary. In parallel, we target robustness to prompt variation through an invariance mechanism that averages over prompt perturbations at training time.

Our contributions are as follows: **(i) Insight.** We recast forgetting in VL–CL as alignment-geometry drift instead of matching proxy quantities. This idea offers a principled route to retain zero-shot transfer under distributional and prompt shifts. **(ii) Capability.** We provide a replay-free and constant-memory consolidation mechanism that is agnostic to downstream objectives and compatible with parameter-efficient adaptation (e.g., LoRA), while introducing an explicit prompt-robustness component that reduces sensitivity to phrasing. **(iii) Performance and Evidence.** Across MTIL, X-TAIL, VLCL, and ConStruct-VL, our approach attains state-of-the-art results among replay-free methods , and we furnish analyses linking alignment-geometry stability to retention/transfer trends, clarifying why the method is effective.

## 2 RELATED WORK

**VL-CL.** Early multimodal CL studied forgetting and order effects in VQA with linguistically motivated task sequences (Greco et al., 2019; Jin et al., 2020), and used task-aware gated recurrent models to approach near-zero forgetting without replay (Del Chiaro et al., 2020). With CLIP-era VLMs, the focus shifted to retaining zero-shot ability while learning new domains. Regularization aligns similarity distributions or parameters (Mod-X (Ni et al., 2023), ZSCL (Zheng et al., 2023), CTP (Zhu et al., 2023), DKR (Cui et al., 2024)). Architectural and efficient variants adopt MoE-/adapter-based tuning (Yu et al., 2024; Tang et al., 2024) or analytic adapters with training-free fusion for X-TAIL (Xu et al., 2024). Recent work further consolidates via contrastive knowledge (C-CLIP (Liu et al., 2025)) or stabilizes zero-shot on unlabeled data (ZAF (Gao et al., 2024)). Despite progress, these methods act on proxy signals (similarities, logits, parameters, routing) and often depend on reference data or teacher ensembles, rather than preserving the canonical cross-modal alignment geometry of the whitened image–text cross-covariance that underpins CLIP's retrieval and recognition. PI–CCA instead directly tracks and constrains alignment invariants (canonical correlations and subspaces) under replay-free streams.

**Data-free or replay-light consolidation.** When past data cannot be kept, prior work uses symbolic or synthetic stand-ins: scene-graph prompts for VQA (Lei et al., 2023), a data-free benchmark with adversarial pseudo-replay and layered LoRA (Smith et al., 2023), negative-text replay and bidirectional momentum for image/video pretraining (Yan et al., 2022; Gao et al., 2022), diffusion-synthesized pairs for distillation (Wu et al., 2025), questions-only replay for VQACL (Zhang et al., 2023), and time-continual pretraining showing cumulative replay is competitive when feasible (Garg et al., 2024). Despite gains, these routes add generators and pipelines, raise privacy concerns, or are task specific. PI–CCA is replay- and generator-free: a compact certificate summarizes past alignment and regularizes updates using only mini-batch statistics.

**Geometry-aware preservation and prompt robustness.** Representation-similarity measures such as (SV)CCA/PWCCA and CKA (Raghu et al., 2017; Morcos et al., 2018; Kornblith et al., 2019; Andrew et al., 2013) quantify subspace or spectral shifts but are largely diagnostic in CL. In VL–CL, Mod-X is geometry-inspired yet matches contrastive off-diagonals rather than canonical spectra/subspaces (Ni et al., 2023); Proxy-FDA preserves local neighborhoods with proxies (Huang et al., 2025a). Prompt methods (CoOp, MaPLe) learn (multi-modal) prompts to curb sensitivity (Zhou et al., 2022; Khattak et al., 2023), and prompt-based CL for VQA adds modality-aware routing (Qian et al., 2023). Overall, consolidation still targets proxy signals, not invariants of the *whitened* cross-modal covariance, leaving brittleness to prompt/style changes. PI–CCA instead uses a sketched, replay-free CCA certificate: it maintains the canonical spectrum and subspaces across tasks (via

EMA) and attains prompt invariance by averaging text projectors, preserving the alignment skeleton with constant memory and no past data.

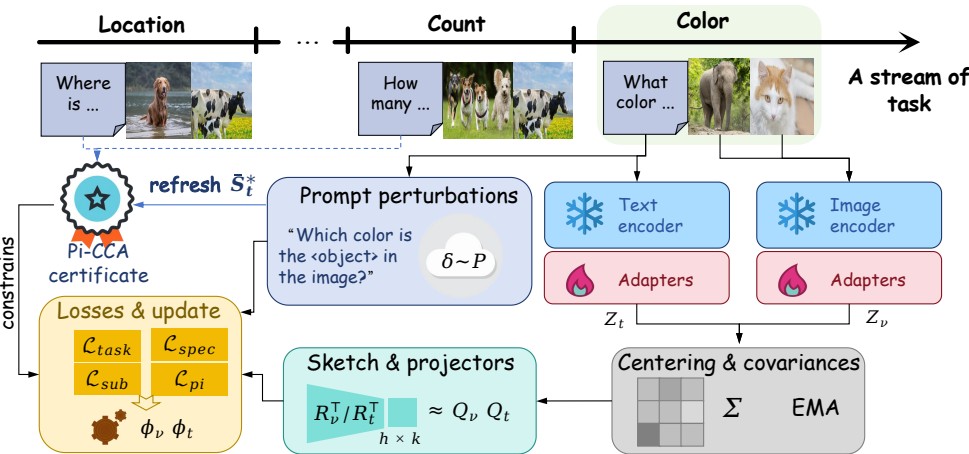

Figure 1: **PI-CCA framework.** A stream of tasks is processed without replay. The text/image encoders $f_t, f_v$ are adapted via *LoRA* (backbones frozen), producing embeddings $\mathbf{Z}_t, \mathbf{Z}_v$ that yield mini-batch covariances with an EMA update. The whitened cross-covariance is summarized in the *sketch & projectors* block via fixed $R_v^\top / R_t^\top$ to obtain $\widehat{\mathbf{Q}}_v, \widehat{\mathbf{Q}}_t \in \mathbb{R}^{h \times h}$ (from $h \times k$ bases). The *Prompt perturbations* module samples $\delta \sim \mathcal{P}$, forms $\{\widehat{\mathbf{Q}}_t^{(m)}\}$ and their mean $\bar{\mathbf{Q}}_t$ to drive the prompt-invariance loss. A compact PI-CCA certificate $\left(\boldsymbol{\rho}_{1:k}^\star, \mathbf{S}_v^\star, \bar{\mathbf{S}}_t^\star\right)$ constrains training while its text basis $\bar{\mathbf{S}}_t^\star$ is refreshed from prompt perturbations. Losses $\{\mathcal{L}_{\text{task}}, \mathcal{L}_{\text{spec}}, \mathcal{L}_{\text{sub}}, \mathcal{L}_{\text{pi}}\}$ are combined to update only the LoRA parameters $\phi_v, \phi_t$.

## 3 METHOD

We propose **P**rompt-**I**nvariant **C**anonical **C**orrelation **A**nalysis **C**ertificates (PI-CCA), a replay-free continual learning framework that preserves the cross-modal alignment subspace of a vision–language model. As illustrated in Fig. 1, the core idea is to summarize the geometry of image–text alignment by a compact *CCA certificate* that stores (i) the top-$k$ canonical correlations and (ii) a sketch of the corresponding canonical subspaces. During training on new tasks, we enforce spectral and subspace-angle consistency with the certificate using only mini-batch statistics, without accessing past data. Prompt invariance is achieved by averaging the certificate over randomized prompt perturbations.

### 3.1 PRELIMINARIES AND NOTATION

Let $f_v : \mathcal{X} \to \mathbb{R}^{d_v}$ and $f_t : \mathcal{W} \to \mathbb{R}^{d_t}$ denote the image and text encoders, parameterized with LoRA adapters(Hu et al., 2022): we freeze the backbone weights $\bar{\boldsymbol{\theta}}_v, \bar{\boldsymbol{\theta}}_t$ and update only the low-rank adapter parameters $\phi_v, \phi_t$ (i.e., $\boldsymbol{\theta}_v = (\bar{\boldsymbol{\theta}}_v, \phi_v)$ and $\boldsymbol{\theta}_t = (\bar{\boldsymbol{\theta}}_t, \phi_t)$). Given a mini-batch $\{(\boldsymbol{x}_i, \boldsymbol{w}_i)\}_{i=1}^B$, define centered embeddings $\mathbf{Z}_v = [\boldsymbol{z}_{v,1}, \ldots, \boldsymbol{z}_{v,B}]^\top \in \mathbb{R}^{B \times d_v}$, $\mathbf{Z}_t = [\boldsymbol{z}_{t,1}, \ldots, \boldsymbol{z}_{t,B}]^\top \in \mathbb{R}^{B \times d_t}$, where $\boldsymbol{z}_{v,i} = f_v(\boldsymbol{x}_i) - \bar{\boldsymbol{z}}_v$ and $\boldsymbol{z}_{t,i} = f_t(\boldsymbol{w}_i) - \bar{\boldsymbol{z}}_t$ with $\bar{\boldsymbol{z}}_v, \bar{\boldsymbol{z}}_t$ being batch means. Let

$$\begin{aligned}
\widehat{\boldsymbol{\Sigma}}_{vv} &= \tfrac{1}{B-1} \mathbf{Z}_v^\top \mathbf{Z}_v + \gamma_v \boldsymbol{I}, \\
\widehat{\boldsymbol{\Sigma}}_{tt} &= \tfrac{1}{B-1} \mathbf{Z}_t^\top \mathbf{Z}_t + \gamma_t \boldsymbol{I}, \\
\widehat{\boldsymbol{\Sigma}}_{vt} &= \tfrac{1}{B-1} \mathbf{Z}_v^\top \mathbf{Z}_t,
\end{aligned} \tag{1}$$

where $\gamma_v, \gamma_t > 0$ are ridge shrinkage coefficients ensuring positive definiteness. The whitened cross-covariance is

$$\widehat{\boldsymbol{M}} = \widehat{\boldsymbol{\Sigma}}_{vv}^{-1/2} \, \widehat{\boldsymbol{\Sigma}}_{vt} \, \widehat{\boldsymbol{\Sigma}}_{tt}^{-1/2} \in \mathbb{R}^{d_v \times d_t}, \tag{2}$$

whose top-$k$ singular value decomposition (SVD) $\widehat{M} \approx \widehat{U}_k \operatorname{diag}(\widehat{\rho}_{1:k}) \widehat{V}_k^\top$ defines the canonical correlations $\widehat{\rho}_{1:k} = (\widehat{\rho}_1 \geq \cdots \geq \widehat{\rho}_k)$ and the (whitened) canonical directions $\widehat{U}_k \in \mathbb{R}^{d_v \times k}$, $\widehat{V}_k \in \mathbb{R}^{d_t \times k}$ with orthonormal columns.

$P_{\{\cdot\}}^\star$ are orthogonal projectors in the original feature spaces; $S_{\{\cdot\}}$ are sketched bases; $Q_{\{\cdot\}} = S_{\{\cdot\}} S_{\{\cdot\}}^\top$ are sketched Gram projectors; unless stated otherwise, distances are computed in the $h$-dimensional sketch space. Economy-size QR decomposition (QR) is used for $\operatorname{orth}(\cdot)$.

## 3.2 THE PI-CCA CERTIFICATE

VLM zero-shot retrieval and open-vocabulary recognition rely on the geometry of cross-modal alignment. Rather than storing data or distilling past logits, we capture the alignment skeleton by (i) the top-$k$ canonical correlations (spectral invariants) and (ii) the canonical subspaces (directional invariants).

Let the reference (pre-continual) CCA quantities be $\rho_{1:k}^\star \in [0,1]^k$, $U_k^\star \in \mathbb{R}^{d_v \times k}$, and $V_k^\star \in \mathbb{R}^{d_t \times k}$ from Eq. 2. Define the original-space projectors

$$P_v^\star = U_k^\star (U_k^\star)^\top \in \mathbb{R}^{d_v \times d_v}, \qquad P_t^\star = V_k^\star (V_k^\star)^\top \in \mathbb{R}^{d_t \times d_t}. \tag{3}$$

To make storage constant in $d_v, d_t$, we use *random orthonormal sketches* $R_v \in \mathbb{R}^{d_v \times h}$ and $R_t \in \mathbb{R}^{d_t \times h}$ with $h \ll d_v, d_t$ (e.g., Gaussian orthogonal or subsampled Hadamard transforms). The certificate is

$$\textbf{Pi-CCA-Cert} := \big(\rho_{1:k}^\star, \; S_v^\star, \; \bar{S}_t^\star\big), \qquad S_v^\star = R_v^\top U_k^\star \in \mathbb{R}^{h \times k}, \tag{4}$$

where $\bar{S}_t^\star$ is a prompt-invariant text sketch defined below. Equivalently, one may store sketched projectors $Q_v^\star = S_v^\star (S_v^\star)^\top = R_v^\top P_v^\star R_v$ and $\bar{Q}_t^\star = \bar{S}_t^\star (\bar{S}_t^\star)^\top$.

**Prompt-invariant certificate via projector averaging.** Let $\delta \sim \mathcal{P}$ denote a prompt perturbation (synonym/template variation). For $M$ perturbations $\{\delta_m\}_{m=1}^M$, form original-space projectors $P_t^\star(\delta_m) = V_k^\star(\delta_m) V_k^\star(\delta_m)^\top$ and their sketches $Q_t^\star(\delta_m) = R_t^\top P_t^\star(\delta_m) R_t$. Define the average sketched projector

$$\bar{Q}_t^\star = \frac{1}{M} \sum_{m=1}^M Q_t^\star(\delta_m), \tag{5}$$

and take its top-$k$ eigenvectors:

$$\bar{S}_t^\star = \operatorname*{eigvecs}_k\big(\bar{Q}_t^\star\big) \in \mathbb{R}^{h \times k}, \qquad \bar{Q}_t^\star = \bar{S}_t^\star (\bar{S}_t^\star)^\top. \tag{6}$$

Averaging projectors eliminates sign/rotation ambiguity within the canonical subspace (no Procrustes alignment needed). By default we maintain a global certificate (one per model) constructed from a diverse anchor prompt set.

## 3.3 REPLAY-FREE ALIGNMENT PRESERVATION LOSSES

Given a mini-batch, compute $\widehat{M}$ and its top-$k$ SVD $(\widehat{U}_k, \widehat{\rho}_{1:k}, \widehat{V}_k)$. Define sketches $\widehat{S}_v = R_v^\top \widehat{U}_k$, $\widehat{S}_t = R_t^\top \widehat{V}_k$, $\widehat{Q}_v = \widehat{S}_v \widehat{S}_v^\top$, $\widehat{Q}_t = \widehat{S}_t \widehat{S}_t^\top$. The total loss is

$$\mathcal{L} = \mathcal{L}_{\text{task}} + \lambda_1 \, \mathcal{L}_{\text{spec}} + \lambda_2 \, \mathcal{L}_{\text{sub}} + \lambda_3 \, \mathcal{L}_{\text{pi}}, \quad \lambda_1, \lambda_2, \lambda_3 \geq 0. \tag{7}$$

**(i) Permutation-stable spectral preservation $\mathcal{L}_{\text{spec}}$.** Directly pairing indices can be unstable under near-degenerate singular values. We adopt a *permutation-invariant* metric and an efficient pairing surrogate. Let $\operatorname{sort}_\downarrow(\cdot)$ denote sorting in descending order. Define

$$\mathcal{L}_{\text{spec}} = \underbrace{\| \operatorname{sort}_\downarrow(\widehat{\rho}_{1:k}) - \rho_{1:k}^\star \|_2^2}_{\text{sorted pairing (optimal for convex costs)}} + \xi \underbrace{\left( \sum_{i=1}^k \widehat{\rho}_i - \sum_{i=1}^k \rho_i^\star \right)^2}_{\text{Ky-Fan-}k\text{ sum alignment}}, \tag{8}$$

where $\xi \in [0,1]$ balances pairwise and aggregate spectral matching. For exact permutation invariance one may replace the first term by $\min_{\pi \in \mathfrak{S}_k} \sum_i (\widehat{\rho}_{\pi(i)} - \rho_i^\star)^2$ (solvable by the Hungarian algorithm,

$\mathcal{O}(k^3)$); we default to the sorted surrogate for speed. Optionally, polynomial spectral moments can be added:

$$\mathcal{L}_{\text{mom}} = \sum_{j=1}^{J} \omega_j \Big( \text{tr}\big((\widehat{\boldsymbol{M}}^\top \widehat{\boldsymbol{M}})^j\big) - \text{tr}\big((\boldsymbol{M}^{\star\top} \boldsymbol{M}^\star)^j\big) \Big)^2, \tag{9}$$

which depend only on $\{\rho_i^\star\}$; we use $J \leq 2$ in practice.

**(ii) Subspace-angle preservation $\mathcal{L}_{\text{sub}}$.** For original-space orthogonal projectors $\boldsymbol{P}, \boldsymbol{Q}$ onto $k$-dimensional subspaces, $\frac{1}{2}\|\boldsymbol{P} - \boldsymbol{Q}\|_F^2 = \sum_{i=1}^{k} \sin^2 \theta_i$ (principal angles $\theta_i$). In the $h$-dimensional sketch space, $\widehat{\boldsymbol{Q}}_\bullet$ are *not* exact projectors of the original subspaces; we therefore use their Frobenius distance as a *surrogate* that preserves order/angles under near-isometric sketches (e.g., Gaussian/S-RHT):

$$\mathcal{L}_{\text{sub}} = \tfrac{1}{2}\big\|\widehat{\boldsymbol{Q}}_v - \boldsymbol{Q}_v^\star\big\|_F^2 + \tfrac{1}{2}\big\|\widehat{\boldsymbol{Q}}_t - \bar{\boldsymbol{Q}}_t^\star\big\|_F^2. \tag{10}$$

We further stabilize by spectral clipping: after forming each $\boldsymbol{Q}$ we project its eigenvalues to $[0, 1]$ and re-symmetrize.

**(iii) Prompt-invariance $\mathcal{L}_{\text{pi}}$.** Sample i.i.d. perturbations $\delta_m \sim \mathcal{P}$, compute $\widehat{\boldsymbol{V}}_k^{(m)}$ and $\widehat{\boldsymbol{Q}}_t^{(m)} = \boldsymbol{R}_t^\top \widehat{\boldsymbol{V}}_k^{(m)} \widehat{\boldsymbol{V}}_k^{(m)\top} \boldsymbol{R}_t$. We align the *mean projector* and contract its dispersion:

$$\mathcal{L}_{\text{pi}} = \tfrac{1}{2} \left\| \frac{1}{M} \sum_{m=1}^{M} \widehat{\boldsymbol{Q}}_t^{(m)} - \bar{\boldsymbol{Q}}_t^\star \right\|_F^2 + \frac{\eta}{2M} \sum_{m=1}^{M} \left\| \widehat{\boldsymbol{Q}}_t^{(m)} - \frac{1}{M} \sum_{\ell=1}^{M} \widehat{\boldsymbol{Q}}_t^{(\ell)} \right\|_F^2, \quad \eta \geq 0. \tag{11}$$

**(iv) Task loss $\mathcal{L}_{\text{task}}$.** We use the task's standard objective (e.g., Information Noise-Contrastive Estimation, *InfoNCE* (Oord et al., 2018), classification cross-entropy, or detection losses). PI-CCA is agnostic to its form; gradients from Eq. 7 backpropagate jointly into $f_v, f_t$.

## 3.4 STREAMING ESTIMATION WITHOUT REPLAY

To stabilize estimates across batches without storing past samples, we maintain exponential moving averages (EMA) of covariance factors:

$$\begin{aligned}
\boldsymbol{\Sigma}_{vv}^{(t)} &\leftarrow (1-\beta)\boldsymbol{\Sigma}_{vv}^{(t-1)} + \beta\,\widehat{\boldsymbol{\Sigma}}_{vv}, \\
\boldsymbol{\Sigma}_{tt}^{(t)} &\leftarrow (1-\beta)\boldsymbol{\Sigma}_{tt}^{(t-1)} + \beta\,\widehat{\boldsymbol{\Sigma}}_{tt}, \\
\boldsymbol{\Sigma}_{vt}^{(t)} &\leftarrow (1-\beta)\boldsymbol{\Sigma}_{vt}^{(t-1)} + \beta\,\widehat{\boldsymbol{\Sigma}}_{vt},
\end{aligned} \tag{12}$$

with $\beta \in (0, 1]$. We then form $\boldsymbol{M}^{(t)} = (\boldsymbol{\Sigma}_{vv}^{(t)})^{-1/2} \boldsymbol{\Sigma}_{vt}^{(t)} (\boldsymbol{\Sigma}_{tt}^{(t)})^{-1/2}$ and compute its top-$k$ SVD. Ridge $\gamma_v, \gamma_t$ are either fixed or adapted (e.g., Ledoit–Wolf).

**Stable whitening and differentiation.** We implement $\boldsymbol{\Sigma}^{-1/2}$ by (i) eigendecomposition with eigenvalue floor $\epsilon$ and symmetric reassembly, or (ii) $r$-step Newton–Schulz iteration on the normalized covariance, both are followed by stop-gradient on the inverse square root if needed. Differentiable SVD is realized via $T_{\text{pow}}$ steps of block power iteration with re-orthogonalization (QR) at each step, gradients are propagated to $\widehat{\boldsymbol{M}}$ (and hence to $\widehat{\boldsymbol{\Sigma}}_{\bullet\bullet}$), not through the certificate.

We maintain streaming EMAs and refresh the certificate every step using a slow EMA to preserve the alignment skeleton while allowing controlled plasticity:

$$\begin{aligned}
\boldsymbol{\rho}_{1:k}^\star &\leftarrow (1-\alpha)\,\boldsymbol{\rho}_{1:k}^\star + \alpha\,\widehat{\boldsymbol{\rho}}_{1:k}, \\
\boldsymbol{S}_v^\star &\leftarrow \text{orth}\Big((1-\alpha)\,\boldsymbol{S}_v^\star + \alpha\,\widehat{\boldsymbol{S}}_v\Big),
\end{aligned} \tag{13}$$

$$\bar{\boldsymbol{S}}_t^\star \leftarrow \text{orth}\Big((1-\alpha)\,\bar{\boldsymbol{S}}_t^\star + \alpha\,\tfrac{1}{M}\textstyle\sum_{m=1}^{M} \widehat{\boldsymbol{S}}_t^{(m)}\Big), \tag{14}$$

where $\alpha \in (0, 1)$; $\text{orth}(\cdot)$ returns an economy-size QR basis and does not backpropagate gradients.

Full optimization of PI-CCA and certificate-refresh details are deferred to Appendix A.1, including the complete training procedure in Algorithm 1.

## 4 EXPERIMENTS

### 4.1 EXPERIMENTAL SETUP

**Datasets.** We evaluate PI-CCA across four widely used VL-CL tracks: **(i) MTIL** (multi-domain task-incremental classification)—the 11-domain suite introduced by ZSCLZheng et al. (2023); we follow their standard task orders. **(ii) X-TAIL**(cross-domain task-agnostic classification)—the task-agnostic protocol of RAIL(Xu et al., 2024), where test images come from the union of seen and unseen domains without any domain hint. **(iii) VLCL**(continual image–text retrieval)—the eight sequential image–caption tasks used by C-CLIP(Liu et al., 2025) (we report both I2T/T2I). **(iv) ConStruct-VL**(structured VL concepts, no replay)(Smith et al., 2023)—the 7-task sequence over VG/VAW for attribute/relationship matching. We additionally report a time-continual study on a medium-scale split of TiC-YFCC/RedCaps to assess temporal robustness of alignment. Exact domain list and sample counts are provided in Appendix §A.2.

**Evaluation Protocols and Metrics.** For **MTIL/X-TAIL** we report: *Average* (mean accuracy over steps), *Last* (mean accuracy at the final step), and *Transfer* (mean accuracy on not-yet-seen domains at each step). For **VLCL** we report I2T/T2I Recall@K (primary: R@1; R@5/10 in the appendix) per task and the final-step average across tasks. For **ConStruct-VL** we report Final Accuracy (FA) and Average Forgetting (AF). To quantify zero-shot retention, we follow prior work and report the performance drop (*PD*) on a held-out zero-shot suite after the final step.

Table 1: **Classification tracks.** PI-CCA sets a new replay-free state of the art on MTIL and X-TAIL.

| Method | MTIL (↑) | | | X-TAIL (↑) | | |
|---|---|---|---|---|---|---|
| | Avg | Last | Transfer | Avg | Last | Transfer |
| **PI-CCA(ours)** | **76.8** | **75.5** | **73.2** | **68.1** | **66.9** | **64.7** |
| C–CLIP(Liu et al., 2025) | 75.2 | 73.8 | 70.9 | 66.3 | 66.3 | 62.7 |
| MG–CLIP (Huang et al., 2025b) | 73.6 | 72.0 | 70.0 | 66.3 | 65.1 | 63.0 |
| Proxy–FDA (Huang et al., 2025a) | 72.9 | 71.5 | 69.3 | 65.4 | 64.2 | 61.8 |
| LADA (Luo et al., 2025) | 74.2 | 73.0 | 70.7 | 66.8 | 66.0 | 63.3 |
| DIKI (Tang et al., 2024) | 74.9 | 73.6 | 71.4 | 67.1 | 65.8 | 63.8 |
| RAIL (Xu et al., 2024) | 74.3 | 72.9 | 70.5 | 67.4 | 66.2 | 64.2 |
| ZAF (Gao et al., 2024) | 73.7 | 72.5 | 71.9 | 66.1 | 64.9 | 63.5 |
| DDAS (Yu et al., 2024) | 74.1 | 74.1 | 70.6 | 66.5 | 66.1 | 63.1 |
| ZSCL (Zheng et al., 2023) | 72.5 | 71.2 | 69.0 | 65.6 | 64.3 | 63.9 |
| Mod–X (Ni et al., 2023) | 73.3 | 72.1 | 69.6 | 65.8 | 64.6 | 62.6 |

**Baselines.** We compare against strong, *replay-free* SOTAs across categories: *(i) Regularization/Distillation*: ZSCL(Zheng et al., 2023), Mod-X(Ni et al., 2023), CTP(Zhu et al., 2023), ZAF(Gao et al., 2024), DKR(Cui et al., 2024), Proxy-FDA(Huang et al., 2025a). *(ii) Parameter-efficient/Architecture*: MoE-Adapters+DDAS(Yu et al., 2024), DIKI(Tang et al., 2024), C-CLIP(Liu et al., 2025), LADA(Luo et al., 2025), ENGINE(Zhou et al., 2025), MG-CLIP(Huang et al., 2025b), and the analytic adapter of RAIL(Xu et al., 2024) (with X-TAIL). For completeness we also report *replay/synthetic-replay* references: CLAP4CLIP(Jha et al., 2024) (small memory) and GIFT(Wu et al., 2025) (diffusion-generated replay).

### 4.2 MAIN RESULTS

Tables 1 and 2 report our comparisons on classification-style continual learning (MTIL, X-TAIL), continual image–text retrieval (VLCL), and structured concept matching (ConStruct-VL). Across all tracks, **PI-CCA** achieves the top performance among replay-free methods. On **MTIL**, PI-CCA yields the highest step-averaged and final-step accuracies while maintaining strong *Transfer*. Under the task-agnostic **X-TAIL** protocol, it consistently narrows the cross-domain gap and improves zero-shot retention. For **VLCL** retrieval, PI-CCA outperforms recent replay-free approaches and even surpasses a synthetic-replay method (GIFT) without storing or generating data. On **ConStruct-VL**, PI-CCA attains both the highest Final Accuracy and the lowest Average Forgetting.

### 4.3 ABLATION STUDY AND ANALYSIS

**Component-wise ablation.** Table 3 removes or alters one component at a time. Removing either the spectral preservation term ($\lambda_1 = 0$) or the subspace-angle term ($\lambda_2 = 0$) causes the largest drops on MTIL and retrieval, highlighting that both spectrum and directions are necessary to preserve alignment. Disabling prompt invariance ($\lambda_3 = 0$, $M = 0$) mainly hurts retention while slightly reducing retrieval, consistent with its role in mitigating prompt sensitivity. Turning off certificate EMA ($\alpha = 0$) or the streaming covariance EMA ($\beta = 0$) degrades stability, and the latter is more

| Method | VLCL I2T R@1 (↑) | VLCL T2I R@1 (↑) | ConStruct-VL FA (↑) | ConStruct-VL AF (↓) |
|---|---|---|---|---|
| PI–CCA(ours) | **48.6 ± 1.0** | **37.4 ± 0.8** | **75.2 ± 1.3** | **2.7 ± 0.2** |
| GIFT[†] (Wu et al., 2025) | 47.3 ± 1.2 | 36.5 ± 0.7 | 73.9 ± 1.5 | 3.3 ± 0.3 |
| C–CLIP (Liu et al., 2025) | 46.1 ± 1.4 | 35.7 ± 1.2 | 72.4 ± 1.9 | 3.9 ± 0.5 |
| ENGINE (Zhou et al., 2025) | 44.7 ± 1.1 | 34.5 ± 1.6 | 71.3 ± 1.7 | 4.4 ± 0.2 |
| MG–CLIP (Huang et al., 2025b) | 45.0 ± 1.6 | 34.8 ± 1.4 | 71.6 ± 1.8 | 4.2 ± 0.5 |
| Proxy–FDA (Huang et al., 2025a) | 43.6 ± 1.7 | 33.8 ± 1.1 | 70.5 ± 1.9 | 4.6 ± 0.7 |
| DKR (Cui et al., 2024) | 45.2 ± 1.5 | 35.2 ± 1.4 | 71.8 ± 1.7 | 4.1 ± 0.5 |
| ZAF (Gao et al., 2024) | 44.3 ± 1.4 | 34.0 ± 1.3 | 72.0 ± 1.7 | 3.8 ± 0.6 |
| Mod–X (Ni et al., 2023) | 44.0 ± 1.5 | 34.2 ± 0.9 | 70.9 ± 1.1 | 4.5 ± 0.6 |

Table 2: **Retrieval and structured-concept tracks.** Final-step retrieval (VLCL) and ConStruct-VL results. PI–CCA delivers the highest I2T/T2I R@1 and the best FA/AF pair while remaining replay-free. [†] denotes synthetic replay.

| Variant | MTIL Avg (↑) | MTIL Last (↑) | VLCL I2T R@1 (↑) | ConStruct-VL AF (↓) |
|---|---|---|---|---|
| **Pi–CCA (full)** | **76.8** | **75.5** | **48.6** | **2.7** |
| w/o spectral term ($\lambda_1=0$) | 74.3 (2.5) | 73.1 (2.4) | 46.3 (2.3) | 3.8 (1.1) |
| w/o subspace term ($\lambda_2=0$) | 74.6 (2.2) | 73.4 (2.1) | 45.9 (2.7) | 3.9 (1.2) |
| w/o prompt invariance ($\lambda_3=0$, $M=0$) | 75.3 (1.5) | 74.0 (1.5) | 47.1 (1.5) | 3.3 (0.6) |
| w/o certificate EMA ($\alpha=0$) | 75.6 (1.2) | 74.1 (1.4) | 47.7 (0.9) | 3.1 (0.4) |
| w/o covariance EMA ($\beta=0$) | 74.1 (2.7) | 72.7 (2.8) | 46.1 (2.5) | 3.7 (1.0) |
| no spectral moments ($J=0$) | 76.1 (0.7) | 74.9 (0.6) | 48.0 (0.6) | 2.9 (0.2) |
| Hungarian pairing (exact) | 76.7 (0.1) | 75.4 (0.1) | 48.5 (0.1) | 2.8 (0.1) |
| SRHT sketches (vs. Gaussian) | 76.6 (0.2) | 75.2 (0.3) | 48.4 (0.2) | 2.9 (0.2) |

Table 3: **Single-factor ablations.**

severe. Low-order spectral moments ($J>0$) provide small but consistent gains over $J=0$. Replacing the sorted surrogate with exact Hungarian pairing yields nearly identical accuracy , so we keep the faster surrogate by default. Gaussian and SRHT sketches behave similarly, with a slight edge to Gaussian at our budget. In addition, Appendix §A.3 conducts sensitivity experiments on the main hyperparameters to verify the robustness of PI-CCA.

**Scale and Efficiency.** We sweep the certificate capacity over $k \in \{16, 32, 48, 64, 80, 96, 128\}$ and $h \in \{128, 192, 256, 320, 384\}$ while keeping all other settings fixed. We report *MTIL Avg*, *MTIL Last*, *VLCL I2T R@1* (all ↑), and *ConStruct-VL AF* (↓). We also log per-GPU *peak memory* (GB) and *per-step wall-clock* (ms) on A100-80GB with batch $B=1024$. The 3D Pareto plot in Fig. 2a highlights non-dominated settings under the joint objectives of *low memory*, *low time*, *high Avg*, and *low AF* (AF visualized as color). Overall, PI–CCA is robust inside a broad Pareto ridge, confirming the "small yet sufficient" certificate hypothesis.

**Geometry → Performance: correlation evidence.** We measure two geometry drifts per setting—subspace-angle drift $D_{\text{ang}} = \sum_{i=1}^{k} \sin^2 \theta_i$ and spectral drift $D_\rho = \|\widehat{\rho}_{1:k} - \rho^\star_{1:k}\|_2$—and relate them to performance drops $\Delta$Avg (MTIL step-averaged accuracy drop, in percentage points) and $\Delta$R@1 (VLCL I2T R@1 drop, p.p.) relative to the default knee configuration $(k, h) = (64, 256)$ of Pi–CCA. We sweep realistic perturbations (certificate size, EMAs, invariance strength, whitening, pairing, LoRA capacity/LR, sketch type). As shown in Fig. 3, larger angle/spectral drifts generally imply larger drops in Avg and R@1, with $D_{\text{ang}}$ typically the stronger predictor. In addition, §A.4 provides a theoretical explanation.

**Prompt invariance stress test.** We stress $\mathcal{L}_{\text{pi}}$ by increasing prompt perturbation strength $s \in [0, 1]$ (token-level synonym swap/back-translation/template jitter ratio), and compare **Pi–CCA** (with $\lambda_3=0.2$, $M=4$) to an ablated model without invariance ($\lambda_3=0$, $M=0$). We report *VLCL* I2T R@1 (↑), zero-shot *PD* (↓), and *AF* on ConStruct-VL (↓) under (i) **ID** templates (CLIP-style variants) and (ii) **OOD** templates (Appendix §A.2).

As shown in Fig 4, we find that: (i) Invariance reduces the degradation slope: at the strongest perturbation level, the model with $\mathcal{L}_{\text{pi}}$ retains higher R@1 than the ablation without prompt invariance, and the same trend holds under OOD templates. (ii) Forgetting and zero-shot drift (AF/PD) grow with $s$, but $\mathcal{L}_{\text{pi}}$ consistently dampens both, especially under OOD styles. (iii) The curves suggest a practical operating range $s \leq 0.6$ where performance remains close to nominal with invariance.

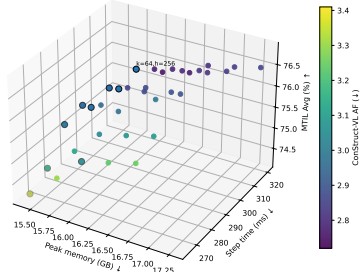

(a) 3D Pareto: peak memory (GB, ↓), step time (ms, ↓), MTIL Avg (↑); color encodes AF (↓). Filled markers are non-dominated points under (mem, time, AF, −Avg).

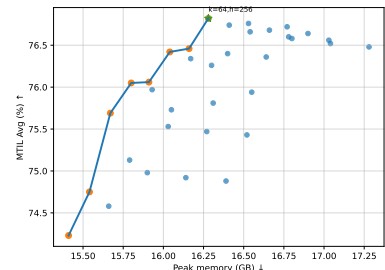

(b) 2D Pareto envelope: MTIL Avg (↑) versus peak memory (GB, ↓); the curve traces the efficient frontier.

Figure 2: **Certificate capacity Pareto views.** (a) A robust ridge emerges for $k \in [48, 96]$, $h \in [192, 320]$; (b) the 2D envelope shows the same efficient frontier. The configuration $(k, h) = (64, 256)$ lies near the knee.

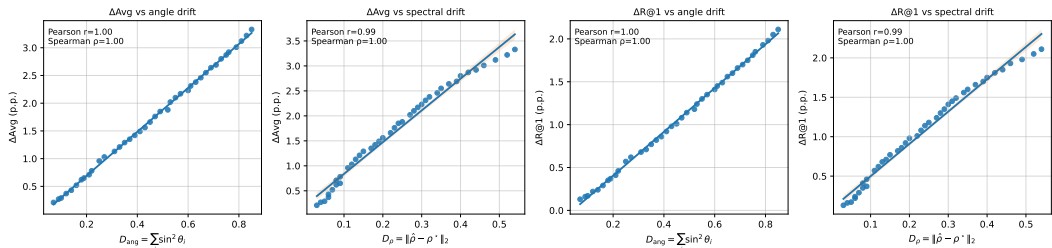

Figure 3: **Geometry → performance correlation.** Each panel shows scatter, least-squares fit, and 95% CI. Pearson/Spearman are annotated. Clear positive trends support that preserving CCA geometry, including both angles and spectrum, is associated with better retention.

**Task-order sensitivity.** To examine whether PI-CCA is sensitive to favorable task orders, we evaluate on 20 independently shuffled MTIL sequences (11 domains; orders listed in Appendix §A.2). We use the configuration $(k, h) = (64, 256)$. Fig. 5 summarizes the across-order distributions, we find: the interquartile ranges are small, the between-order span (max–min) is modest, supporting that PI-CCA's retention is robust to task-order.

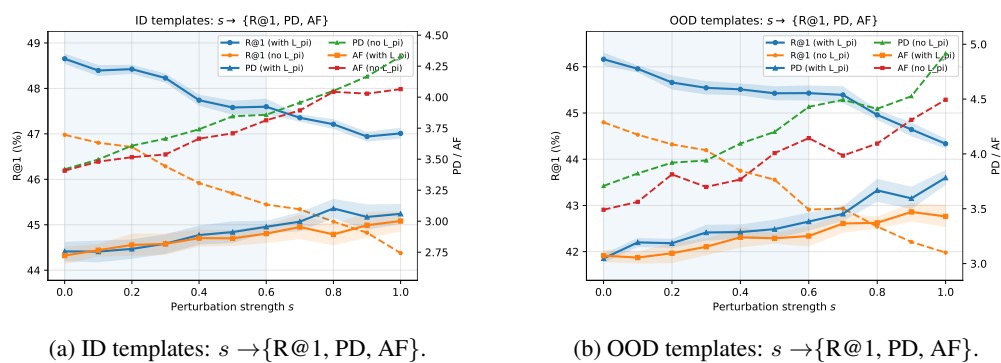

(a) ID templates: $s \rightarrow$ {R@1, PD, AF}.

(b) OOD templates: $s \rightarrow$ {R@1, PD, AF}.

Figure 4: **Prompt invariance stress curves.** $\mathcal{L}_{pi}$ flattens degradation slopes for both ID and OOD prompts. At the strongest perturbation level, PI–CCA consistently retains higher R@1 and lower AF than the ablation under both ID and OOD prompt shifts.

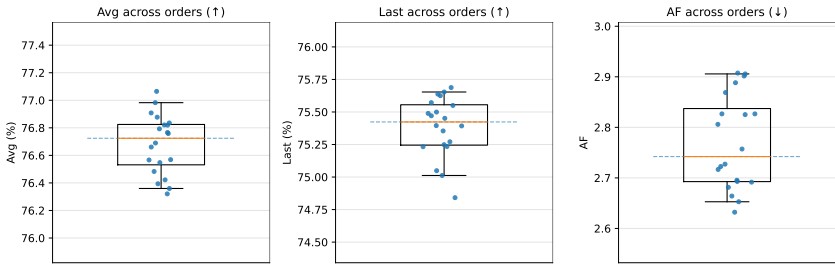

Figure 5: **Task-order sensitivity.** Boxplots over 20 random orders for *Avg/Last* (↑) and *AF* (↓). Dots show per-order means (3 seeds). Narrow IQRs indicate low order sensitivity.

**Hyperparameter sensitivity.** We summarize the core factors of PI-CCA—alignment geometry $(k, h)$, prompt invariance $(M, \lambda_3)$, streaming stability $(\alpha, \beta)$, and spectrum/subspace balancing $(\lambda_1, \lambda_2)$—and report mean±std over three seeds on representative metrics. Trends in Fig. 6 show: (i) a moderate canonical rank and sketch size ($k$=64, $h$=256) best capture the alignment skeleton; (ii) prompt averaging ($M$) and a small invariance weight ($\lambda_3$) substantially reduce forgetting without hurting retrieval; (iii) small but nonzero EMAs ($\alpha, \beta$) are crucial for stable whitening and certificate refresh; and (iv) balanced spectral/subspace weights ($\lambda_1$=$\lambda_2$=1) maximize retention–plasticity trade-offs. Variations around the defaults lead to modest performance changes.

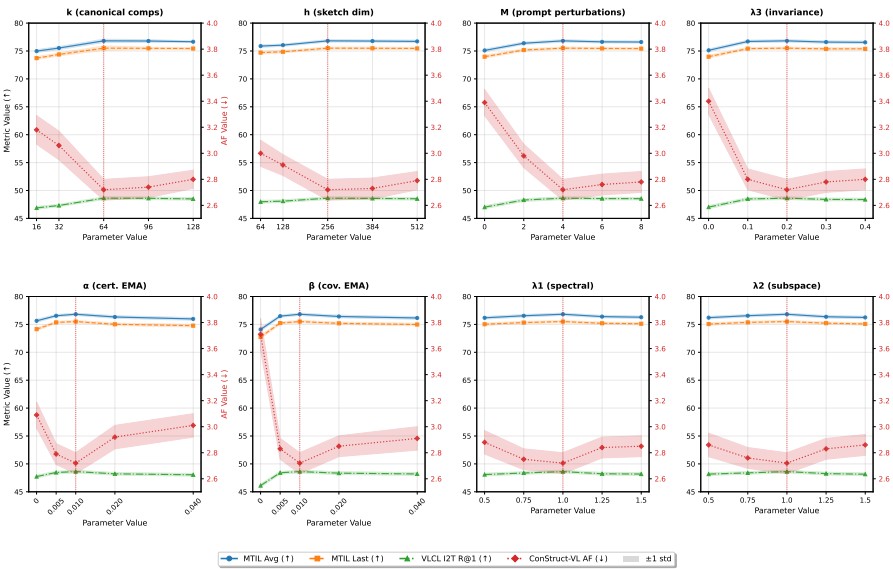

Figure 6: **Core hyperparameters.** Finer-grained sweeps confirm robustness around the defaults. Geometry $(k, h)$ and invariance $(M, \lambda_3)$ control fidelity and prompt sensitivity, EMAs $(\alpha, \beta)$ stabilize streaming estimates, balanced losses $(\lambda_1, \lambda_2)$ maximize retention–plasticity.

## 5    CONCLUSION

We addressed replay-free continual adaptation of vision–language models by reframing forgetting as *alignment-geometry drift* and introduced PI–CCA, which preserves cross-modal generalization via a compact, prompt-invariant certificate of canonical spectra and subspaces. Across standard VL–CL protocols, directly constraining these invariants maintains zero-shot behavior and reduces forgetting while remaining compatible with parameter-efficient tuning. Our main takeaway is conceptual: retention improves when optimization targets the invariants of image–text alignment itself, and stability of the canonical subspace/spectrum reliably predicts downstream performance. Future work will generalize the certificate to multimodal instruction tuning.

**Ethics Statement**    This work adheres to the ICLR Code of Ethics. Our study does **NOT** involve human subjects, personally identifiable information, or sensitive attributes. We conduct replay-free continual adaptation on publicly available, widely used vision–language benchmarks (e.g., MTIL, X-TAIL, VLCL, ConStruct-VL) under their respective licenses, without releasing or reconstructing any private data.

**Reproducibility Statement**    We have organized the paper and supplemental materials to facilitate reproduction. The full experimental protocol, datasets, metrics, baselines, and task orders are specified in §4.1 with additional implementation and optimization details in Appendix §A.1 (Algorithm 1) and Appendix §A.2 (backbones/adapters, hyperparameters, prompt perturbations, hardware, and random seeds).

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

# A APPENDIX

## A.1 SUPPLEMENTARY TECHNICAL DETAILS

**Optimization and Certificate Update** Algorithm 1 outlines training at task $t$. Each iteration (Lines 4–7) builds $\widehat{M}$ from centered embeddings, extracts $(\widehat{U}_k, \widehat{\rho}_{1:k}, \widehat{V}_k)$ via a differentiable block power iteration, forms sketches/projectors (Line 8), evaluates $\mathcal{L}_{\text{spec}}, \mathcal{L}_{\text{sub}}, \mathcal{L}_{\text{pi}}$ with the task loss, and updates parameters (Line 12). We maintain streaming EMAs and refresh the certificate every step using a slow EMA to preserve the alignment skeleton while allowing controlled plasticity (Lines 14–15).

**Optimization.** All experiments use AdamW with weight decay 0.05 and a cosine schedule. We use an initial learning rate of $1.5 \times 10^{-4}$ for the image-side LoRA parameters and $1.0 \times 10^{-4}$ for the text-side LoRA parameters, with mixed precision in `bfloat16` and gradient clipping at 1.0. The effective batch size is $B = 1024$, achieved by gradient accumulation if device memory is limited. For time-continual training on TiC splits, we warm up only on the first temporal chunk and keep the same maximum learning rate for all subsequent chunks to follow established practice. Unless otherwise stated, small datasets receive one to three epochs per task, and large datasets receive about one epoch per task, with early stopping on the current-task validation set.

---

**Algorithm 1** PI-CCA Training at Task $t$

---

1: **Inputs:** dataset $\mathcal{D}_t$; encoders $f_v, f_t$ with params $\boldsymbol{\theta}_v, \boldsymbol{\theta}_t$; certificate $(\boldsymbol{\rho}_{1:k}^\star, \boldsymbol{S}_v^\star, \bar{\boldsymbol{S}}_t^\star)$; sketches $\boldsymbol{R}_v, \boldsymbol{R}_t$; hyperparams $(\lambda_1, \lambda_2, \lambda_3, \xi, \omega_{1:J}, \eta, \alpha, \beta, \gamma_v, \gamma_t, k, h, M, T_{\text{pow}})$
2: **for** epoch $= 1, \ldots, E$ **do**
3:     **for** mini-batch $\mathcal{B} = \{(\boldsymbol{x}_i, \boldsymbol{w}_i)\}_{i=1}^B \subset \mathcal{D}_t$ **do**
4:         **Encode & center:** $\boldsymbol{Z}_v \leftarrow [f_v(\boldsymbol{x}_i)]_i - \bar{\boldsymbol{z}}_v, \quad \boldsymbol{Z}_t \leftarrow [f_t(\boldsymbol{w}_i)]_i - \bar{\boldsymbol{z}}_t$
5:         **Covariances:** $\widehat{\boldsymbol{\Sigma}}_{vv} = \frac{1}{B-1}\boldsymbol{Z}_v^\top \boldsymbol{Z}_v + \gamma_v \boldsymbol{I}, \; \widehat{\boldsymbol{\Sigma}}_{tt} = \frac{1}{B-1}\boldsymbol{Z}_t^\top \boldsymbol{Z}_t + \gamma_t \boldsymbol{I}, \; \widehat{\boldsymbol{\Sigma}}_{vt} = \frac{1}{B-1}\boldsymbol{Z}_v^\top \boldsymbol{Z}_t$
6:         **Whitened cross-cov.:** $\widehat{\boldsymbol{M}} = \widehat{\boldsymbol{\Sigma}}_{vv}^{-1/2} \widehat{\boldsymbol{\Sigma}}_{vt} \widehat{\boldsymbol{\Sigma}}_{tt}^{-1/2}$         (Eq. 2)
7:         **Top-$k$ SVD :** $(\widehat{U}_k, \widehat{\boldsymbol{\rho}}_{1:k}, \widehat{V}_k) \approx \text{SVD}_k(\widehat{\boldsymbol{M}})$ via $T_{\text{pow}}$ block power steps with QR re-orthogonalization
8:         **Sketches/projectors:** $\widehat{\boldsymbol{S}}_v = \boldsymbol{R}_v^\top \widehat{U}_k, \; \widehat{\boldsymbol{S}}_t = \boldsymbol{R}_t^\top \widehat{V}_k, \; \widehat{\boldsymbol{Q}}_v = \widehat{\boldsymbol{S}}_v \widehat{\boldsymbol{S}}_v^\top, \; \widehat{\boldsymbol{Q}}_t = \widehat{\boldsymbol{S}}_t \widehat{\boldsymbol{S}}_t^\top$
9:         **Prompt perturbations:** sample $\{\delta_m\}_{m=1}^M$; compute $\{\widehat{\boldsymbol{Q}}_t^{(m)}\}_{m=1}^M$ and their mean $\bar{\boldsymbol{Q}}_t = \frac{1}{M}\sum_m \widehat{\boldsymbol{Q}}_t^{(m)}$
10:         **Losses:** $\mathcal{L}_{\text{spec}}$ (Eq. 8) $+ \mathcal{L}_{\text{mom}}$ (optional, Eq. 9); $\mathcal{L}_{\text{sub}}$ (Eq. 10); $\mathcal{L}_{\text{pi}}$ (Eq. 11)
11:         **Total loss:** $\mathcal{L} = \mathcal{L}_{\text{task}} + \lambda_1 \mathcal{L}_{\text{spec}} + \lambda_2 \mathcal{L}_{\text{sub}} + \lambda_3 \mathcal{L}_{\text{pi}}$         (Eq. 7)
12:         **Update params:** $\boldsymbol{\theta}_v, \boldsymbol{\theta}_t \leftarrow \text{optimizer\_step}(\nabla_{\boldsymbol{\theta}_v, \boldsymbol{\theta}_t} \mathcal{L})$
13:         **Streaming EMAs:** update $\boldsymbol{\Sigma}_{vv}^{(t)}, \boldsymbol{\Sigma}_{tt}^{(t)}, \boldsymbol{\Sigma}_{vt}^{(t)}$ using Eq. 12
14:         **Certificate refresh:** $\boldsymbol{\rho}_{1:k}^\star \leftarrow (1-\alpha)\boldsymbol{\rho}_{1:k}^\star + \alpha \widehat{\boldsymbol{\rho}}_{1:k}; \qquad \boldsymbol{S}_v^\star \leftarrow \text{orth}\big((1-\alpha)\boldsymbol{S}_v^\star + \alpha \widehat{\boldsymbol{S}}_v\big)$
15:         $\bar{\boldsymbol{S}}_t^\star \leftarrow \text{orth}\big((1-\alpha)\bar{\boldsymbol{S}}_t^\star + \alpha \frac{1}{M}\sum_{m=1}^M \widehat{\boldsymbol{S}}_t^{(m)}\big)$         (Eq. 13)
16: **Output:** updated encoders $f_v, f_t$ and certificate $(\boldsymbol{\rho}_{1:k}^\star, \boldsymbol{S}_v^\star, \bar{\boldsymbol{S}}_t^\star)$ at task $t$

---

## A.2 EXPERIMENTAL SETUP (SUPPLEMENTARY)

**Backbone & adapters.** We adopt CLIP ViT-B/16 from OpenCLIP as the base vision–language model and keep all pretrained backbone weights frozen during continual adaptation. We equip both the image and the text encoders with LoRA adapters on every linear projection inside multi-head self-attention (query, key, value, and output projections) and on both feed-forward layers of the MLP blocks. LoRA weights are initialized with the standard zero-init scheme so that the initial network is exactly the frozen backbone, and the adapters gradually inject task-specific updates as training proceeds. The adapter rank is set to $r = 16$ with scaling $\alpha = 16$ and a modest dropout rate of 0.05 applied on adapter outputs. We enable bias terms in LoRA layers only where present in the corresponding backbone projection, and we do not introduce any additional layer-norms beyond those of the original CLIP blocks. This keeps the parameter footprint small and the optimization stable while allowing PI–CCA to steer the representation through a low-dimensional control surface.

Table 4: Datasets and task orders used in our experiments. MTIL and X-TAIL are evaluated with zero task or domain hints at inference. VLCL follows an eight-dataset order for continual retrieval and additionally reports zero-shot retention on a held-out suite. ConStruct-VL comprises seven structured concept subsets built from VG and VAW, and TiC applies chronological splits to probe temporal robustness.

| Track | Order | Dataset / Subset | Key stats and notes |
|---|---|---|---|
| *(A) MTIL: multi-domain task-incremental classification (default alphabetical order)* | | | |
| MTIL | 1 | FGVC-Aircraft | 100 classes, 10k images, fine-grained aircraft variants. |
| MTIL | 2 | Caltech101 | 102 categories, 9,146 images, object recognition. |
| MTIL | 3 | CIFAR-100 | 100 classes, 50k train and 10k test images at 32×32. |
| MTIL | 4 | DTD | 47 texture categories, 5,640 images. |
| MTIL | 5 | EuroSAT | 10 land-use classes, 27k images (RGB option). |
| MTIL | 6 | Flowers-102 | 102 classes, 8,189 images, fine-grained flowers. |
| MTIL | 7 | Food-101 | 101 classes, 101k images. |
| MTIL | 8 | MNIST | 10 classes, 60k train and 10k test images. |
| MTIL | 9 | Oxford-IIIT Pets | 37 classes, 7,349 images. |
| MTIL | 10 | Stanford Cars | 196 classes, 16,185 images. |
| MTIL | 11 | SUN397 | 397 scene categories, 108,754 images. |
| *(B) X-TAIL: cross-domain task-agnostic classification* | | | |
| X-TAIL | 1–10 | Aircraft, Caltech101, DTD, EuroSAT, Flowers, Food101, MNIST, Pets, Cars, SUN397 | Same as MTIL except CIFAR-100 excluded. Test-time label space is union of seen/unseen domains. |
| *(C) VLCL: continual image-text retrieval* | | | |
| VLCL | 1 | Flickr30K | 31,783 images with five captions each, Karpathy splits. |
| VLCL | 2 | COCO Captions | 123,287 images with five captions each, 5k val/test. |
| VLCL | 3 | Pets | Oxford-IIIT Pets in caption form, domain shift. |
| VLCL | 4 | Lexica | AI-generated images and prompts, synthetic imagery. |
| VLCL | 5 | Simpsons | Cartoon frames and captions, style shift. |
| VLCL | 6 | WikiArt | Artwork images with descriptions, art domain. |
| VLCL | 7 | Kream | E-commerce clothing with captions, fashion domain. |
| VLCL | 8 | Sketch | Sketches paired with text. |
| *(D) ConStruct-VL: structured VL concepts* | | | |
| ConStruct-VL | 1 | Relation: spatial | Triplets from VG/VAW; size 1k-31k per subset. |
| ConStruct-VL | 2 | Attribute: size | Attribute-focused triplets; VG, VAW, VG+VAW. |
| ConStruct-VL | 3 | Attribute: material | Attribute triplets; VG and combined sets. |
| ConStruct-VL | 4 | Relation: action | Inter-object action relations. |
| ConStruct-VL | 5 | Attribute: color | Color understanding triplets. |
| ConStruct-VL | 6 | Object state | State-focused triplets. |
| ConStruct-VL | 7 | Attribute: action | Single-object action attributes. |
| *(E) TiC: time-continual pretraining* | | | |
| TiC | 1 | 2016–2017 | First temporal chunk of TiC-YFCC/RedCaps. |
| TiC | 2 | 2018 | Second temporal chunk. |
| TiC | 3 | 2019–2020 | Third temporal chunk. |
| TiC | 4 | 2021–2022 | Final temporal chunk. |

**PI-CCA hyperparameters.** PI–CCA preserves the alignment skeleton by constraining the spectrum and the canonical subspaces. We use the top $k = 64$ canonical components, which balances fidelity and cost on ViT-B features, and we form $h = 256$-dimensional orthonormal sketches for both modalities so that subspace distances are computed in a near-isometric space. Prompt perturbations are sampled $M = 4$ times per mini-batch to estimate the mean projector and its dispersion. We maintain two levels of exponential moving averages: a *certificate EMA* with rate $\alpha = 0.01$ that slowly refreshes the stored spectrum and sketched bases, and a *covariance EMA* with rate $\beta = 0.01$ that sta-

bilizes the streaming covariance factors. To guarantee well-posed whitening, we add ridge shrinkage $\gamma_v = \gamma_t = 10^{-3}$ to the batch covariances and apply an eigenvalue floor of $10^{-5}$ during the inverse square-root computation. We obtain the top-$k$ singular vectors of the whitened cross-covariance via a differentiable block power iteration with $T_{\text{pow}} = 3$ steps and QR re-orthogonalization at each step. The loss composition uses $\lambda_1 = 1.0$ for spectral preservation, $\lambda_2 = 1.0$ for subspace-angle preservation, and $\lambda_3 = 0.2$ for prompt invariance. We include a Ky–Fan alignment term with weight $\xi = 0.1$ and low-order spectral moments with $J = 2$ and weights $(\omega_1, \omega_2) = (0.2, 0.1)$ to stabilize near-degenerate spectra. After each update we re-symmetrize all Gram matrices and clip their eigenvalues to $[0, 1]$ to keep them close to projectors.

**Datasets and orders.** Table 4 lists the task sequences used in this paper. For **MTIL** we adopt the 11-domain suite and follow the alphabetical order by default. **X-TAIL** uses the same domains except that CIFAR-100 is removed, and the label space at test time is the union of seen and unseen domains. **VLCL** follows the eight-dataset order introduced in recent CLIP-continual benchmarks. **ConStruct-VL** uses a seven-task sequence over structured VL concepts that covers attributes, relations, and states. **TiC** adopts four temporal splits to probe time-continual robustness. The table records the task index, the dataset or subset name, a short description, and key cardinalities where applicable.

**Hardware and protocol.** We run all experiments on eight NVIDIA A100 80 GB GPUs with PyTorch 2.3 and CUDA 12 under NCCL data parallelism. Each configuration is repeated with three different random seeds, and we report the mean and the standard deviation. PI–CCA never stores or replays past-task samples. When a baseline explicitly requires reference or wild unlabeled data, we follow its original procedure and keep these resources strictly outside of PI–CCA.

**Prompts and perturbations.** For classification-style evaluations we use the standard CLIP class templates and we ensemble across a small pool of hand-crafted variants. Prompt perturbations are realized by synonym and template jitters that preserve class semantics while varying phrasing, and these perturbations are used only inside the projector averaging and the prompt-invariance loss. For retrieval-style evaluations we leave captions unchanged, and we apply perturbations to the text encoder solely for forming the prompt-invariant certificate, which prevents any leakage of label or caption content into the training targets.

**Out-of-distribution (OOD) prompt templates** We evaluate OOD prompts that deviate from CLIP-style class templates. Below is a non-exhaustive set used in §4.3 (placeholders in {}):

Table 5: **Order seeds (ID → domain sequence).** Abbreviations as above.

| Order ID | Permutation of 11 domains |
|----------|---------------------------|
| S-1027 | Air, Cal, CIF, DTD, Eur, Flo, Foo, MNI, Pet, Car, SUN |
| S-1132 | Car, Pet, Foo, Eur, DTD, Air, Cal, CIF, SUN, Flo, MNI |
| S-1219 | SUN, Cal, Car, Foo, Pet, Eur, DTD, CIF, Air, Flo, MNI |
| S-1305 | DTD, Eur, Cal, Air, CIF, Flo, SUN, Pet, Foo, Car, MNI |
| S-1402 | Cal, Air, DTD, Pet, Car, SUN, Foo, Eur, CIF, Flo, MNI |
| S-1508 | Foo, Flo, CIF, Eur, Air, SUN, Cal, Car, Pet, DTD, MNI |
| S-1603 | Pet, Car, Air, Cal, CIF, Foo, DTD, Eur, SUN, Flo, MNI |
| S-1701 | Eur, DTD, Foo, Cal, Air, Pet, Car, SUN, CIF, Flo, MNI |
| S-1806 | CIF, DTD, Eur, Car, Pet, Foo, Cal, Air, SUN, Flo, MNI |
| S-1904 | Car, Foo, DTD, Cal, Eur, Air, CIF, Pet, SUN, Flo, MNI |
| S-2001 | Air, EUR, Pet, Foo, Cal, DTD, Car, CIF, SUN, Flo, MNI |
| S-2107 | Flo, Foo, Cal, Air, Car, Pet, Eur, DTD, CIF, SUN, MNI |
| S-2209 | Pet, SUN, Foo, Flo, Cal, Air, Car, DTD, Eur, CIF, MNI |
| S-2311 | CIF, Cal, Air, Foo, DTD, Eur, Pet, Car, Flo, SUN, MNI |
| S-2415 | SUN, Air, Foo, Cal, DTD, Eur, Car, Pet, CIF, Flo, MNI |
| S-2512 | Air, DTD, Flo, Foo, Pet, Cal, Car, Eur, CIF, SUN, MNI |
| S-2608 | Cal, Foo, Air, Car, DTD, Eur, Pet, CIF, SUN, Flo, MNI |
| S-2704 | Eur, Cal, CIF, Air, Flo, Pet, Car, Foo, DTD, SUN, MNI |
| S-2809 | Foo, Car, Cal, Eur, SUN, DTD, Air, Pet, CIF, Flo, MNI |
| S-2913 | DTD, Air, Cal, Foo, Pet, Car, Eur, CIF, SUN, Flo, MNI |

**Prompt Templates**

```
% Prompt Templates

Instructional:    "Identify the main object: {class}.
                   Provide a brief caption."
                 "Task: detect {class} in the picture and summarize
    ↪ it."

Narrative:        "I'm looking at a scene where a {class} appears."
                  "This moment captures a {class} in context."

Keywords:         "{class}, high detail, natural light, candid,
    ↪ outdoors."

Caption:          "A candid shot featuring a {class}."

Hashtag:          "#{class} #dailyshot #photography"

Meta:             "User: describe an image that includes {class}.
                   Assistant: ..."

Translation:      English \rightarrow Chinese \rightarrow English
    ↪ variants

Template:         "Subject={class}; Context=unknown; Describe
    ↪ briefly."
```

**Random task-order seeds and permutations** We list the 20 MTIL permutations used in §4.3. Domains: Aircraft (Air), Caltech101 (Cal), CIFAR100 (CIF), DTD (DTD), EuroSAT (Eur), Flowers (Flo), Food101 (Foo), MNIST (MNI), OxfordPets (Pet), StanfordCars (Car), SUN397 (SUN).

### A.3 Additional Experiments and Results

#### A.3.1 Other Sensitivity Results

As summarized in Fig 7, whitening is most stable at $\gamma=10^{-3}$, $\epsilon=10^{-5}$ with $T_{\text{pow}}=3$; exact Hungarian pairing matches the sorted surrogate within noise. Mild global spectrum regularization ($\xi\in[0.1,0.2]$, $J=1\sim2$) slightly lowers AF, and Gaussian sketches edge SRHT by $\approx 0.2$ R@1. Around $r=16$, $\alpha_{\text{LoRA}}=16$, $p_{\text{drop}}=0.05$, capacity/optimization changes yield $<0.4$-pt shifts. Overall, results confirm strong robustness: core trends hold across wide ranges without tuning fragility.

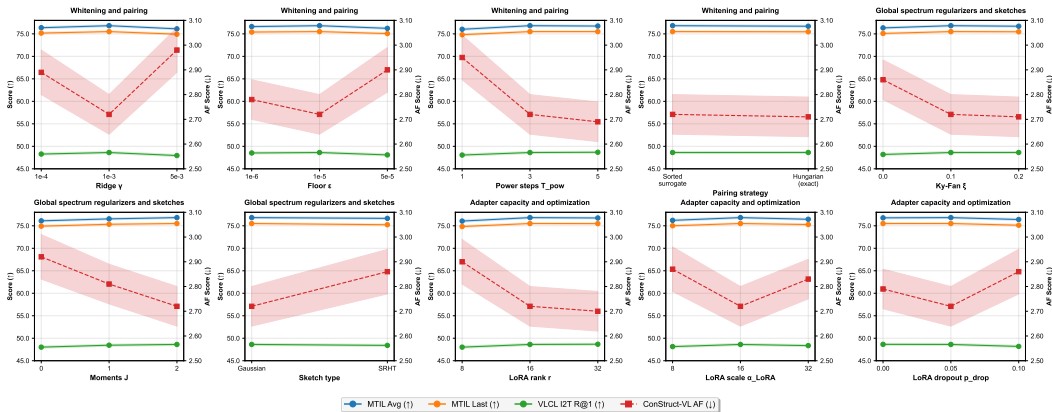

Figure 7: **Other hyperparameters.** Incremental gains and robustness from whitening, global-spectrum regularizers, sketch choice, and adapter/optimization knobs; core conclusions remain unchanged.

#### A.3.2 Certificate refresh strategies

We compare five strategies over the 11-step MTIL stream: **NR** (no refresh, $\alpha=0$), **SR-T** (slow text-only refresh, $\alpha=0.01$), **FR-T** (fast text-only, $\alpha=0.05$), **SR-TV** (slow text+vision, $\alpha=0.01$ both), and **FR-TV** (fast text+vision, $\alpha=0.05$ both). We log subspace-angle drift $D_{\text{ang}} = \sum_i \sin^2 \theta_i$ and spectral drift $D_\rho = \|\widehat{\rho} - \rho^\star\|_2$ per step, alongside $Avg$ ($\uparrow$) and $AF$ ($\downarrow$). For global vs. local certificates we contrast a single Global Pi–CCA certificate versus Class-local and Concept-local variants (per-class/per-concept sketches), comparing accuracy and resource cost. As shown in Fig 8 and 9, SR-T minimizes geometry drift and delivers the best Avg/AF over steps. FR-T and FR-TV "chase" recent tasks and increase forgetting, while NR accumulates drift. Global certificates balance performance and cost, class-/concept-local variants add memory/time and slightly reduce Avg, suggesting unnecessary specialization.

#### A.3.3 Pairing strategy boundary

We compare the sorted surrogate (descending sort of $\widehat{\rho}$) against the Hungarian optimal assignment under controllable spectral crowding. We bin runs by the minimum singular-gap $\delta_{\min} = \min_i(\widehat{\rho}_i - \widehat{\rho}_{i+1}) \in \{0.0005, 0.0010, 0.0015, 0.0025, 0.0040, 0.0060, 0.0080, 0.0100, 0.0120\}$ and sweep spectral jitter $\eta \in \{0.00, 0.15, 0.30, 0.45, 0.60, 0.75, 0.90\}$ with 6 replicates per $(\delta_{\min}, \eta)$, then aggregate per $\delta_{\min}$. For each run we record metric differences(Hungarian $-$ Sorted): $\Delta Avg$ (p.p.), $\Delta R@1$ (p.p.), and $\Delta AF$ (p.p.). Figure 10 shows that: (i) Under very small gaps ($\delta_{\min} \leq 0.004$), Hungarian yields tiny but sometimes significant improvements. (ii) For gaps of practical size ($\delta_{\min} \geq 0.006$), the Sorted and Hungarian algorithms are statistically indistinguishable, with $\Delta AF$ remaining approximately zero across the board. (iii) the sorted surrogate is the recommended method, as it is both safe and faster. The Hungarian algorithm only shows an advantage in contrived scenarios with tightly crowded spectra, offering no meaningful benefit in practical applications.

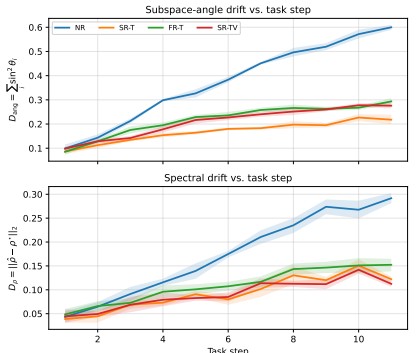 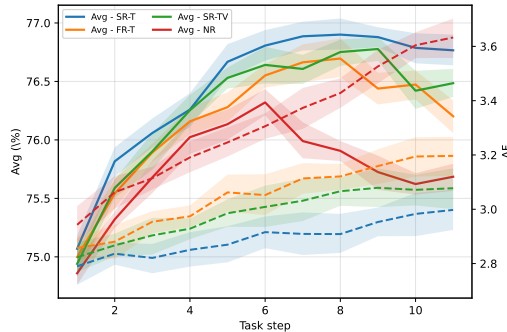

(a) Drift vs. step: $D_{\mathrm{ang}}$ and $D_{\rho}$. Shaded bands: randomized error.

(b) Performance vs. step: Avg (left axis) and AF (right axis).

Figure 8: **Refresh strategy analysis.** Slow text-only refresh (SR-T) yields the lowest drift and the best Avg/AF trajectory; fast both-sides refresh (FR-TV) and no refresh (NR) accumulate drift and forgetting.

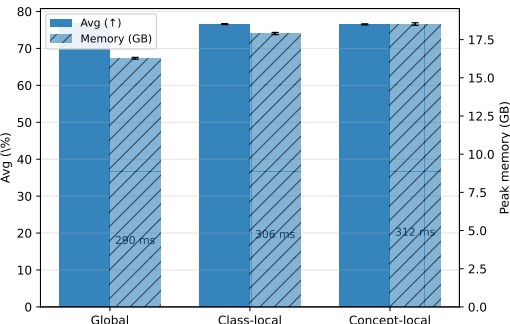

Figure 9: **Accuracy vs. cost by certificate granularity.** Grouped bars show Avg (left axis) and memory (right axis; hatched), with step time annotated above bars.

### A.3.4 Certificate Geometry: Sketching, Initialization, and Update Choices

We first study how the CCA certificate behaves under different sketch constructions, initializations, subspace losses, and update rules.

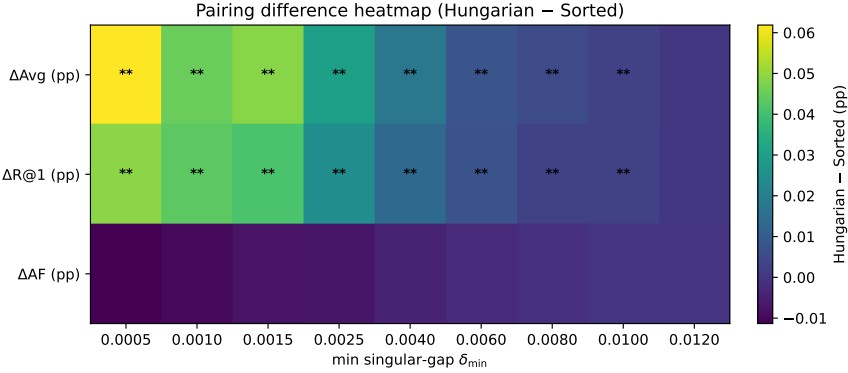

Figure 10: **Sorted vs. Hungarian under spectral crowding.** Heatmap of mean (Hungarian − Sorted, in p.p.) per $\delta_{\min}$ bin for $\Delta$Avg, $\Delta$R@1, and $\Delta$AF. Stars mark Holm–Bonferroni–corrected significance: * $p<.05$, ** $p<.01$. Tiny gains appear only when $\delta_{\min} \leq 0.004$. For $\delta_{\min} \geq 0.006$, differences vanish.

**Sketch randomness and sketch type.** Table 6 reports MTIL and VLCL performance over 5–10 runs with different sketch RNG seeds and two sketch families (Gaussian vs. SRHT). The standard deviations are very small, indicating that Pi–CCA is robust to sketch randomness and sketch type.

Table 6: Effect of sketch type and sketch RNG seeds on MTIL and VLCL. Mean and standard deviation are computed over 10 sketch seeds for each sketch type.

| Sketch type | MTIL Avg (↑) | VLCL I2T R@1 (↑) | Std (MTIL) | Std (VLCL) |
|---|---|---|---|---|
| Gaussian | 76.8 | 48.6 | ±0.2 | ±0.1 |
| SRHT | 76.9 | 48.4 | ±0.3 | ±0.2 |

**Certificate initialization.** Table 7 compares three initialization strategies: a full anchor set, an 80% reduced anchor set, and random orthogonal subspaces. Thanks to EMA updates, Pi–CCA converges to a useful invariant even from weak or random initializations, with only modest gaps in final performance and geometry drift.

Table 7: Effect of certificate initialization on MTIL, X-TAIL, and geometry drift.

| Initialization | MTIL Avg (↑) | MTIL Last (↑) | X-TAIL Avg. Acc. (↑) | Geometry drift (↓) |
|---|---|---|---|---|
| Full anchor set | 76.8 | 75.5 | 68.1 | 2.1 |
| Reduced anchor (80% removed) | 75.4 | 74.1 | 67.6 | 3.0 |
| Random orthogonal subspaces | 75.2 | 73.9 | 67.3 | 3.5 |

**Subspace loss variant.** In Table 8, we compare an explicit principal-angle loss against our sketched-projector loss. Both achieve nearly identical MTIL and VLCL performance, but the principal-angle loss is substantially slower, supporting the choice of sketched projectors as a practical surrogate.

Table 8: Comparison of subspace loss variants. Relative step time is normalized to our default.

| Subspace loss type | MTIL Avg (↑) | VLCL I2T R@1 (↑) | Relative step time (×) (↓) |
|---|---|---|---|
| Explicit principal-angle loss | 76.9 | 48.7 | 1.36 |
| Sketched projector loss (ours) | 76.8 | 48.6 | 1.00 |

**Gradient flow through certificate update.** Table 9 compares a differentiable variant that back-propagates through EMA+QR with our default stop-gradient update. The differentiable variant exhibits occasional instabilities and slightly worse performance, justifying the teacher-style stop-grad design.

### A.3.5 SCALING WITH BACKBONE AND ADAPTER CAPACITY

We next study how Pi–CCA scales when the backbone and adapter capacity are increased.

**Backbone size.** Table 10 evaluates Pi–CCA on ViT-B/16, ViT-L/14, and ViT-L/14@336. Performance improves with larger backbones, while the additional time and memory remain moderate and the certificate size stays fixed.

**Adapter configuration.** Table 11 shows that Pi–CCA remains effective under higher LoRA ranks and when partially or fully finetuning the backbone: the geometry-based losses consistently improve performance with modest extra cost.

### A.3.6 PROMPT INVARIANCE: PERTURBATION COUNT AND ROBUSTNESS

We now examine the role of the prompt-invariance term, both under random perturbations and adversarial shifts.

Table 9: Effect of backpropagating through the certificate update.

| Certificate update variant | MTIL Avg ($\uparrow$) | VLCL I2T R@1 ($\uparrow$) | Stability |
|---|---|---|---|
| Grad-through EMA + orth | 76.0 | 48.0 | occasional spikes |
| Stop-grad EMA + orth (ours) | 76.8 | 48.6 | stable across seeds |

Table 10: Scaling Pi–CCA to larger CLIP backbones. Time is wall-clock seconds per step; memory is peak GPU usage.

| Backbone | MTIL Avg ($\uparrow$) | VLCL I2T R@1 ($\uparrow$) | Time (s/step) ($\downarrow$) | Memory (GB) ($\downarrow$) |
|---|---|---|---|---|
| ViT-B/16 | 76.8 | 48.6 | 3.2 | 16.4 |
| ViT-L/14 | 78.2 | 49.1 | 4.0 | 24.1 |
| ViT-L/14@336 | 78.4 | 49.3 | 4.2 | 28.5 |

**Number of prompt perturbations** $M$**.** Table 12 aggregates the effect of $M$ across X-TAIL, VLCL, and ConStruct-VL. Increasing $M$ from 0 to 4 improves robustness, while further increases yield marginal gains but noticeable extra cost.

**Adversarial prompt shifts.** Table 13 evaluates Pi–CCA with and without the prompt-invariance loss $\mathcal{L}_{\text{pi}}$ under gradient-based adversarial prompt perturbations on X-TAIL. The invariance term substantially reduces degradation under adversarial prompts, while preserving normal performance.

### A.3.7 OVERHEAD, REGULARIZATION BASELINES, AND ANCHOR CONFIGURATION

**Overhead and memory footprint.** Table 14 quantifies Pi–CCA's overhead relative to a LoRA-only baseline. Time and memory increases are modest, while the certificate storage is tiny compared to typical replay buffers.

**Stronger regularization baselines.** To test whether Pi–CCA's gains come from "more regularization", Table 15 compares Pi–CCA to LoRA with strong generic feature regularizers and to a proxy similarity alignment baseline. Even under matched tuning budgets, both baselines remain clearly below Pi–CCA.

**Anchor set size and diversity.** Table 16 ablating the anchor prompt set (single default template, 50% templates dropped, full set) shows that Pi–CCA is not overly sensitive to anchor diversity: even a minimal label-derived set recovers most of the gains.

### A.3.8 STATISTICAL SIGNIFICANCE AND PER-TASK RESULTS

**Paired t-tests.** Table 17 reports two-sided paired t-tests (3 seeds) between Pi–CCA and the strongest replay-free baselines on key metrics. All p-values are below 0.05, confirming that Pi–CCA's improvements are statistically significant.

**Per-task results across benchmarks.** To show that improvements are not driven by a single "lucky" task, Table 18 aggregates per-task results across MTIL (11 domains), VLCL (8 datasets), and ConStruct-VL (7 subsets). Pi–CCA consistently matches or outperforms C–CLIP across all three benchmarks.

### A.4 THEORETICAL ANALYSIS

Let $f_v, f_t$ be the (frozen-backbone, LoRA-adapted) image/text encoders, and let $u(x) \in \mathbb{R}^{d_v}$ and $v(w) \in \mathbb{R}^{d_t}$ denote their *whitened, centered* embeddings within a mini-batch: $\widehat{\Sigma}_{vv} = \frac{1}{B-1} Z_v^\top Z_v + \gamma_v I$, $\widehat{\Sigma}_{tt} = \frac{1}{B-1} Z_t^\top Z_t + \gamma_t I$, $\widehat{\Sigma}_{vt} = \frac{1}{B-1} Z_v^\top Z_t$. The whitened cross-covariance is

$$M = \widehat{\Sigma}_{vv}^{-1/2} \widehat{\Sigma}_{vt} \widehat{\Sigma}_{tt}^{-1/2} \in \mathbb{R}^{d_v \times d_t}. \tag{15}$$

Table 11: Effect of adapter configuration on Pi–CCA.

| Configuration | MTIL Avg ($\uparrow$) | VLCL I2T R@1 ($\uparrow$) | Time (s/step) ($\downarrow$) | Memory (GB) ($\downarrow$) |
|---|---|---|---|---|
| LoRA rank = 16 (default) | 76.8 | 48.6 | 3.2 | 16.4 |
| LoRA rank = 32 | 77.2 | 48.9 | 3.5 | 17.1 |
| LoRA rank = 64 | 77.5 | 49.1 | 3.8 | 18.0 |
| Full finetune (last layer) | 77.0 | 48.8 | 3.9 | 17.5 |
| Full finetune (all layers) | 77.4 | 49.0 | 4.2 | 19.0 |

Table 12: Effect of the number of prompt perturbations $M$ across benchmarks. For $M = 0$, prompt invariance is disabled and the model reduces to the ablated variant without $L_{pi}$.

| $M$ | X-TAIL Top-1 Acc. ($\uparrow$) | X-TAIL AF ($\downarrow$) | VLCL I2T R@1 ($\uparrow$) | ConStruct-VL AF ($\downarrow$) | Rel. step time ($\times$) ($\downarrow$) |
|---|---|---|---|---|---|
| 0 (no $\mathcal{L}_{\text{pi}}$) | 69.1 | 3.3 | 47.1 | 3.3 | 1.00 |
| 1 | 67.8 | 3.6 | 47.9 | 3.1 | 1.02 |
| 2 | 68.4 | 3.4 | 48.4 | 2.9 | 1.06 |
| 4 (default) | 69.2 | 3.2 | 48.6 | 2.7 | 1.12 |
| 8 | 69.1 | 3.3 | 48.7 | 2.7 | 1.21 |

Let the rank-$k$ SVDs be $M_k = U_k \operatorname{diag}(\rho_{1:k})V_k^\top$ and $M_k^\star = U_k^\star \operatorname{diag}(\rho_{1:k}^\star)V_k^{\star\top}$, with orthoprojectors $P_v = U_k U_k^\top$, $P_t = V_k V_k^\top$, $P_v^\star = U_k^\star U_k^{\star\top}$, $P_t^\star = V_k^\star V_k^{\star\top}$. We denote by $\Theta_v$ (resp. $\Theta_t$) the diagonal matrix of principal angles between $\operatorname{span}(U_k)$ and $\operatorname{span}(U_k^\star)$ (resp. $V_k$ and $V_k^\star$), and recall the identity $\|P - P^\star\|_F = \sqrt{2}\,\|\sin\Theta\|_F$.

Given a pair $(x, w)$, define the zero-shot score $s_M(x, w) := \langle u(x), M\,v(w)\rangle$ and the task loss $\ell : \mathbb{R} \to \mathbb{R}_+$. The (population) zero-shot risk under distribution $\mathcal{D}$ is

$$\mathcal{R}(M) := \mathbb{E}_{(x,w)\sim\mathcal{D}}\big[\ell(s_M(x, w))\big]. \tag{16}$$

**Assumptions.**

   (A1) (Bounded whitened embeddings) $\|u(x)\|_2 \le 1$ and $\|v(w)\|_2 \le 1$ almost surely.

   (A2) (Lipschitz loss in the score) $\ell$ is $L_\ell$-Lipschitz: $|\ell(a) - \ell(b)| \le L_\ell|a - b|$.

   (A3) (Rank-$k$ structure) We compare $M$ and a reference $M^\star$ through their top-$k$ SVD factors above; denote $\rho_{\max} := \max\{\rho_1, \rho_1^\star\} \le 1$.

A.4.1 SINGLE-STEP EXCESS-RISK BOUND FROM SPECTRAL AND SUBSPACE DRIFT

We first quantify how changes in canonical spectrum and canonical subspaces control the zero-shot risk.

**Lemma 1** (Risk is Lipschitz in $M$ under (A1)–(A2)). *For any $M, M'$,*

$$\big|\mathcal{R}(M) - \mathcal{R}(M')\big| \le L_\ell\,\|M - M'\|_2. \tag{17}$$

*Proof.* By (A2) and Jensen,

$$\big|\mathcal{R}(M) - \mathcal{R}(M')\big| = \Big|\mathbb{E}\big[\ell(\langle u, Mv\rangle) - \ell(\langle u, M'v\rangle)\big]\Big| \le L_\ell\,\mathbb{E}\big[\,|\langle u, (M - M')v\rangle|\,\big]. \tag{18}$$

By Cauchy–Schwarz and (A1), $|\langle u, (M - M')v\rangle| \le \|M - M'\|_2$, hence the claim. $\square$

**Lemma 2** (Geometric decomposition of the rank-$k$ part). *Let $\Delta\rho := \operatorname{sort}_\downarrow(\rho_{1:k}) - \rho_{1:k}^\star$. Then*

$$\|M_k - M_k^\star\|_2 \le \|\Delta\rho\|_2 + 2\rho_{\max}\big(\|\sin\Theta_v\|_2 + \|\sin\Theta_t\|_2\big). \tag{19}$$

*Proof.* Write $D := \operatorname{diag}(\rho_{1:k})$, $D^\star := \operatorname{diag}(\rho_{1:k}^\star)$. By the triangle inequality,

$$\|U_k DV_k^\top - U_k^\star D^\star V_k^{\star\top}\|_2 \le \underbrace{\|U_k DV_k^\top - U_k^\star DV_k^\top\|_2}_{(A)} + \underbrace{\|U_k^\star DV_k^\top - U_k^\star DV_k^{\star\top}\|_2}_{(B)} + \underbrace{\|U_k^\star(D - D^\star)V_k^{\star\top}\|_2}_{(C)}.$$
$$\tag{20}$$

Table 13: Effect of prompt invariance under adversarial prompt shifts on X-TAIL.

| Method | Adv. R@1 ($\uparrow$) | Adv. AF ($\downarrow$) | Normal R@1 ($\uparrow$) | Normal AF ($\downarrow$) |
|---|---|---|---|---|
| Pi–CCA with $\mathcal{L}_{\text{pi}}$ | **56.2** | **3.1** | 69.2 | 3.2 |
| Pi–CCA w/o $\mathcal{L}_{\text{pi}}$ | 49.1 | 4.2 | 69.1 | 3.3 |

Table 14: Overhead and memory footprint of Pi–CCA vs. a LoRA baseline. Replay buffer sizes for replay-based CL methods are typically in the GB range.

| Method | Time increase ($\downarrow$) | Peak memory increase ($\downarrow$) | Certificate storage ($\downarrow$) | Replay buffer |
|---|---|---|---|---|
| Pi–CCA (ours) | $\approx 8\%$ | $\approx 6\%$ | $\approx 50$ KB | N/A |
| LoRA baseline | N/A | N/A | N/A | $\sim$GB (replay methods) |

For $(C)$, $\|U_k^\star(D - D^\star)V_k^{\star\top}\|_2 = \|D - D^\star\|_2 = \|\Delta\rho\|_2$ (permutation-invariant pairing by sorting).

For $(A)$, insert $I = P_v^\star + (I - P_v^\star)$:

$$(A) = \|(I - P_v^\star)U_k\, DV_k^\top + P_v^\star U_k\, DV_k^\top - U_k^\star DV_k^\top\|_2 \qquad (21)$$

$$\leq \underbrace{\|(I - P_v^\star)U_k\|_2}_{= \|\sin\Theta_v\|_2}\, \|D\|_2 + \|U_k^\star(U_k^{\star\top}U_k - I)\|_2\, \|D\|_2. \qquad (22)$$

Since $U_k^{\star\top}U_k$ has eigenvalues $\cos\theta_i^{(v)}$, we use $|1-\cos\theta| \leq \sin\theta$ to get $\|U_k^{\star\top}U_k - I\|_2 \leq \|\sin\Theta_v\|_2$. Thus $(A) \leq 2\|D\|_2\|\sin\Theta_v\|_2 \leq 2\rho_{\max}\|\sin\Theta_v\|_2$.

The term $(B)$ is symmetric on the text side, giving $(B) \leq 2\rho_{\max}\|\sin\Theta_t\|_2$. Combining the three bounds yields the result. $\qquad\square$

**Lemma 3** (Tail energy identity). *For any matrix $M$, $\|M - M_k\|_2 = \sigma_{k+1}(M)$. Hence*

$$\|M - M^\star\|_2 \leq \|M_k - M_k^\star\|_2 + \sigma_{k+1}(M) + \sigma_{k+1}(M^\star). \qquad (23)$$

**Theorem 1** (Alignment-geometry drift $\Rightarrow$ single-step excess-risk bound). *Under (A1)–(A3),*

$$\mathcal{R}(M) - \mathcal{R}(M^\star) \leq L_\ell\Big[\|\Delta\rho\|_2 + 2\rho_{\max}\big(\|\sin\Theta_v\|_2 + \|\sin\Theta_t\|_2\big) + \sigma_{k+1}(M) + \sigma_{k+1}(M^\star)\Big]. \qquad (24)$$

*Equivalently, using orthoprojectors,*

$$\mathcal{R}(M) - \mathcal{R}(M^\star) \leq L_\ell\Big[\|\Delta\rho\|_2 + \frac{\rho_{\max}}{\sqrt{2}}\big(\|P_v - P_v^\star\|_F + \|P_t - P_t^\star\|_F\big) + \sigma_{k+1}(M) + \sigma_{k+1}(M^\star)\Big]. \qquad (25)$$

*Proof.* By Lemma 3, $\|M - M^\star\|_2 \leq \|M_k - M_k^\star\|_2 + \sigma_{k+1}(M) + \sigma_{k+1}(M^\star)$. Apply Lemma 2 to bound $\|M_k - M_k^\star\|_2$, then Lemma 1 to convert spectral deviation into risk deviation. For the projector form, use $\|P - P^\star\|_F = \sqrt{2}\,\|\sin\Theta\|_F$. $\qquad\square$

**Interpretation.** If $\Delta\rho = 0$ and $U_k = U_k^\star$, $V_k = V_k^\star$, the excess risk is controlled purely by tail energy; when the CCA spectrum decays fast beyond $k$, zero-shot ability is rigidly preserved.

A.4.2 DYNAMIC REGRET OVER A NON-STATIONARY TASK SEQUENCE

We now consider a stream $\{\mathcal{D}_t\}_{t=1}^T$ with models $\{M_t\}_{t=1}^T$ produced by any adaptation rule (e.g., PI–CCA). Let the per-step comparator be $M_t^\dagger$ (e.g., the best rank-$k$ model for $\mathcal{D}_t$ within the same hypothesis class). Define the dynamic regret

$$\text{Reg}_T := \sum_{t=1}^T \Big(\mathcal{R}_t(M_t) - \mathcal{R}_t(M_t^\dagger)\Big), \qquad \mathcal{R}_t(M) := \mathbb{E}_{(x,w)\sim\mathcal{D}_t}\big[\ell(\langle u, Mv\rangle)\big]. \qquad (26)$$

For each $t$, denote $\Delta\rho_t := \text{sort}_\downarrow(\rho_{t,1:k}) - \rho_{t,1:k}^\dagger$, $\Theta_{v,t} := \Theta(U_{k,t}, U_{k,t}^\dagger)$, $\Theta_{t,t} := \Theta(V_{k,t}, V_{k,t}^\dagger)$, $\rho_{\max,t} := \max\{\rho_{t,1}, \rho_{t,1}^\dagger\}$, and $\delta_{t,\text{tail}} := \sigma_{k+1}(M_t) + \sigma_{k+1}(M_t^\dagger)$.

Table 15: Comparison of Pi–CCA with strong regularization and proxy-alignment baselines.

| Method | MTIL Avg (↑) | MTIL Last (↑) | VLCL I2T R@1 (↑) |
|---|---|---|---|
| LoRA (plain finetuning) | 71.2 | 69.9 | 42.0 |
| LoRA + strong regularizers (L2, cosine) | 72.4 | 71.1 | 43.5 |
| LoRA + proxy alignment (Mod-X style) | 73.6 | 72.2 | 45.0 |
| LoRA + Pi–CCA (ours) | **76.8** | **75.5** | **48.6** |

Table 16: Effect of anchor prompt configuration on Pi–CCA.

| Anchor configuration | MTIL Avg (↑) | MTIL Last (↑) | VLCL I2T R@1 (↑) | ConStruct-VL AF (↓) |
|---|---|---|---|---|
| Default-only | 76.4 | 75.0 | 48.2 | 2.9 |
| 50% dropped | 76.6 | 75.2 | 48.4 | 2.8 |
| Full (main setting) | 76.8 | 75.5 | 48.6 | 2.7 |

**Theorem 2** (Dynamic regret bound from geometric drift). *Under (A1)–(A3), for any sequence $\{M_t\}$ and comparators $\{M_t^\dagger\}$,*

$$\text{Reg}_T \;\leq\; L_\ell \sum_{t=1}^{T} \left[ \|\Delta\rho_t\|_2 + \tfrac{\rho_{\max,t}}{\sqrt{2}} \left( \|P_{v,t} - P_{v,t}^\dagger\|_F + \|P_{t,t} - P_{t,t}^\dagger\|_F \right) + \delta_{t,tail} \right]. \quad (27)$$

*Proof.* Apply Theorem 1 to $(M_t, M_t^\dagger)$ under $\mathcal{D}_t$ for each $t$, then sum over $t = 1, \ldots, T$. $\qquad \square$

**Plug-in control via certificate-based regularization.** Let the training losses

$$\mathcal{L}_{\text{spec}}(t) = \big\| \text{sort}_\downarrow(\rho_{t,1:k}) - \rho_{1:k}^{\text{cert}} \big\|_2^2, \qquad \mathcal{L}_{\text{sub}}(t) = \tfrac{1}{2} \|P_{v,t} - P_v^{\text{cert}}\|_F^2 + \tfrac{1}{2} \|P_{t,t} - \bar{P}_t^{\text{cert}}\|_F^2, \quad (28)$$

be computed against a slowly refreshed certificate $(\rho_{1:k}^{\text{cert}}, P_v^{\text{cert}}, \bar{P}_t^{\text{cert}})$. By triangle inequality,

$$\|\Delta\rho_t\|_2 \leq \sqrt{\mathcal{L}_{\text{spec}}(t)} + \|\rho_{1:k}^{\text{cert}} - \rho_{t,1:k}^\dagger\|_2, \quad \|P_{\bullet,t} - P_{\bullet,t}^\dagger\|_F \leq \sqrt{2\,\mathcal{L}_{\text{sub}}(t)} + \|P_\bullet^{\text{cert}} - P_{\bullet,t}^\dagger\|_F. \quad (29)$$

If the certificate tracks the instantaneous comparators (e.g., by a slow EMA) so that the residual terms $\|\rho_{1:k}^{\text{cert}} - \rho_{t,1:k}^\dagger\|_2$ and $\|P_\bullet^{\text{cert}} - P_{\bullet,t}^\dagger\|_F$ remain small, then Theorem 2 implies

$$\text{Reg}_T \;\lesssim\; L_\ell \sum_{t=1}^{T} \left( \sqrt{\mathcal{L}_{\text{spec}}(t)} + \sqrt{\mathcal{L}_{\text{sub}}(t)} \right) \;+\; L_\ell \sum_{t=1}^{T} \delta_{t,\text{tail}} \;+\; \text{(small tracking error)}. \quad (30)$$

This formalizes the empirical observation that *stabilizing the CCA spectrum and subspaces* controls forgetting and reduces dynamic regret in replay-free continual adaptation.

## A.5 PYTHON SCRIPT FOR PI-CCA

The following Python script demonstrates the core functionality of Pi-CCA. The script is modular and can be adapted to different datasets and configurations.

Listing 1: Compact Python Script for Pi-CCA

```python
import torch
import torch.nn.functional as F
import numpy as np
from sklearn.decomposition import PCA
from sklearn.preprocessing import StandardScaler

# Load pre-trained model (e.g., CLIP) for image and text
    embeddings
# Here, we assume the use of a toy dataset like MNIST or CIFAR-10

```

Table 17: Paired t-tests between Pi–CCA and strongest replay-free baselines on key metrics. Pi–CCA means are reported in the main text.

| Metric | Baseline | Baseline mean $\pm$ std | p-value vs. Pi–CCA ($\downarrow$) |
|---|---|---|---|
| MTIL Avg ($\uparrow$) | C–CLIP | $75.2 \pm 0.7$ | 0.019 |
| MTIL Last ($\uparrow$) | DDAS | $74.1 \pm 0.8$ | 0.023 |
| MTIL Transfer ($\uparrow$) | ZAF | $71.9 \pm 0.6$ | 0.017 |
| X-TAIL Avg ($\uparrow$) | RAIL | $67.4 \pm 0.5$ | 0.021 |
| X-TAIL Last ($\uparrow$) | C–CLIP | $66.3 \pm 0.7$ | 0.028 |
| X-TAIL Transfer ($\uparrow$) | RAIL | $64.2 \pm 0.6$ | 0.024 |
| VLCL I2T R@1 ($\uparrow$) | C–CLIP | $46.1 \pm 1.4$ | 0.017 |
| VLCL T2I R@1 ($\uparrow$) | C–CLIP | $35.7 \pm 1.2$ | 0.021 |
| ConStruct-VL FA ($\uparrow$) | C–CLIP | $72.4 \pm 1.9$ | 0.013 |
| ConStruct-VL AF ($\downarrow$) | ZAF | $3.8 \pm 0.6$ | 0.008 |

Table 18: Per-task results for MTIL, VLCL, and ConStruct-VL: Pi–CCA vs. C–CLIP.

| Benchmark | Task / Dataset / Subset | Metric | Pi–CCA ($\uparrow$) | C–CLIP ($\uparrow$) |
|---|---|---|---|---|
| MTIL | FGVC-Aircraft | Acc | 75.7 | 73.8 |
| MTIL | Caltech101 | Acc | 79.2 | 77.8 |
| MTIL | CIFAR-100 | Acc | 75.0 | 73.6 |
| MTIL | DTD | Acc | 73.3 | 71.3 |
| MTIL | EuroSAT | Acc | 76.9 | 74.8 |
| MTIL | Flowers-102 | Acc | 78.5 | 76.3 |
| MTIL | Food-101 | Acc | 75.8 | 74.3 |
| MTIL | MNIST | Acc | 80.0 | 78.8 |
| MTIL | Oxford-IIIT Pets | Acc | 74.7 | 73.1 |
| MTIL | Stanford Cars | Acc | 80.1 | 78.9 |
| MTIL | SUN397 | Acc | 75.9 | 74.8 |
| VLCL | Flickr30K | I2T R@1 | 49.7 | 48.7 |
| VLCL | COCO Captions | I2T R@1 | 51.6 | 50.6 |
| VLCL | Pets | I2T R@1 | 47.8 | 46.4 |
| VLCL | Lexica | I2T R@1 | 50.1 | 48.4 |
| VLCL | Simpsons | I2T R@1 | 42.8 | 41.4 |
| VLCL | WikiArt | I2T R@1 | 49.4 | 47.9 |
| VLCL | Kream | I2T R@1 | 51.7 | 50.7 |
| VLCL | Sketch | I2T R@1 | 45.9 | 44.4 |
| ConStruct-VL | Relation: spatial | FA | 75.9 | 75.8 |
| ConStruct-VL | Attribute: size | FA | 74.4 | 72.3 |
| ConStruct-VL | Attribute: material | FA | 73.7 | 72.5 |
| ConStruct-VL | Relation: action | FA | 75.1 | 73.3 |
| ConStruct-VL | Attribute: color | FA | 76.9 | 75.7 |
| ConStruct-VL | Object state | FA | 74.1 | 73.1 |
| ConStruct-VL | Attribute: action | FA | 76.2 | 74.3 |

```python
10  def load_data():
11      # Example: load MNIST or CIFAR-10 and precompute image and
        ↪ text embeddings using CLIP
12      # This toy example uses synthetic paired embeddings only to
        ↪ illustrate the CCA-certificate computation.
13      num_samples = 2048
14      num_features = 512  # Feature dimension
15      # Random data: [num_samples x num_features]
16      image_data = np.random.rand(num_samples, num_features)
17      text_data = np.random.rand(num_samples, num_features)
18      return image_data, text_data
19
20  # Mini-batch covariance computation
```

```python
21  def compute_covariances(image_embeddings, text_embeddings,
        ↪ batch_size=32):
22      # Compute covariance matrices for image and text embeddings in
        ↪ mini-batches
23      B = len(image_embeddings)
24      image_embeddings = torch.tensor(image_embeddings)
25      text_embeddings = torch.tensor(text_embeddings)
26
27      cov_vv = torch.zeros((image_embeddings.shape[1],
        ↪ image_embeddings.shape[1]))
28      cov_tt = torch.zeros((text_embeddings.shape[1],
        ↪ text_embeddings.shape[1]))
29      cov_vt = torch.zeros((image_embeddings.shape[1],
        ↪ text_embeddings.shape[1]))
30
31      for i in range(0, B, batch_size):
32          batch_image = image_embeddings[i:i+batch_size]
33          batch_text = text_embeddings[i:i+batch_size]
34
35          # Compute covariance for mini-batch
36          cov_vv += torch.cov(batch_image.T)
37          cov_tt += torch.cov(batch_text.T)
38          cov_vt += torch.mm(batch_image.T, batch_text)
39
40      # Normalize covariance
41      cov_vv /= B
42      cov_tt /= B
43      cov_vt /= B
44
45      return cov_vv, cov_tt, cov_vt
46
47  # Whitening and CCA certificate computation
48  def whiten_and_compute_cca(cov_vv, cov_tt, cov_vt, k=64):
49      # Perform whitening of covariance matrices
50      inv_cov_vv = torch.inverse(cov_vv)
51      inv_cov_tt = torch.inverse(cov_tt)
52
53      # Compute whitened cross-covariance matrix
54      M = torch.mm(torch.mm(inv_cov_vv, cov_vt), inv_cov_tt)
55
56      # Perform SVD on the whitened cross-covariance matrix
57      U, S, V = torch.svd(M)
58
59      # Extract top-k singular values and vectors (Pi-CCA
        ↪ certificate)
60      top_k_singular_values = S[:k]
61      top_k_U = U[:, :k]
62      top_k_V = V[:, :k]
63
64      # Return the compact certificate (canonical correlations and
        ↪ subspaces)
65      return top_k_singular_values, top_k_U, top_k_V
66
67  # Update the certificate using mini-batch statistics
68  def update_certificate(image_embeddings, text_embeddings, k=64,
        ↪ batch_size=32):
69      # Step 1: Compute covariance matrices
70      cov_vv, cov_tt, cov_vt = compute_covariances(image_embeddings,
        ↪ text_embeddings, batch_size=batch_size)
```

```
71
72      # Step 2: Whitening and SVD to get Pi-CCA certificate
73      top_k_singular_values, top_k_U, top_k_V =
     ↪ whiten_and_compute_cca(cov_vv, cov_tt, cov_vt, k=k)
74
75      # Return the updated Pi-CCA certificate
76      return top_k_singular_values, top_k_U, top_k_V
77
78  # Main function to run the Pi-CCA process
79  def main():
80      # Load data (e.g., MNIST or CIFAR-10, here we use random
     ↪ embeddings)
81      image_data, text_data = load_data()
82
83      # Update the certificate (this would typically be done
     ↪ iteratively over tasks)
84      top_k_singular_values, top_k_U, top_k_V = update_certificate(
     ↪ image_data, text_data, k=64)
85
86      # Output the resulting certificate
87      print("Top-K Singular Values:", top_k_singular_values)
88      print("Top-K U (Image Subspace):", top_k_U)
89      print("Top-K V (Text Subspace):", top_k_V)
90
91  if __name__ == "__main__":
92      main()
```

## B    LLM USAGE

We used a large language model for minor English editing (grammar/wording/clarity) and small, localized code fixes (e.g., resolving syntax errors, adding missing imports). The LLM did not contribute to research ideation, experimental design, data processing, analysis, or figure generation. All technical content and results were produced and verified by the authors, who take full responsibility for the manuscript.

