# OpenReview forum: "Pi-CCA: Prompt-Invariant CCA Certificates for Replay-Free Continual Multimodal Learning"
_ICLR.cc/2026/Conference — ICLR 2026 Poster_

### Official Review · Reviewer_Z4s6 · 2025-10-28

**Soundness:** 2
**Presentation:** 3
**Contribution:** 2
**Rating:** 6
**Confidence:** 2

**Summary:**

The paper proposes a new method of replay-free continual learning for CLIP-like vision-language models. The core idea is to retain the statistics of feature correlations called CCA certificates, consisting of top-k singular values and the corresponding subspace in the feature space during continual learning. Additionally, they propose to apply low-rank random projections to the CCA certificates, resulting in the quantities called PI-CCA. Also they propose to marginalize these quantities by perturbations in prompts. By leveraging the PI-CCA, it is expected that the proposed continual learning may be more efficient than explicitly leveraging actual data. Experiments show that the proposed method is superior to the existing replay-free methods in average.

**Strengths:**

- The paper is clearly written and easy to follow.
- The core idea of retaining correlation statistics seems simple and easy to implement.
- Experimental results show the superiority of the proposed method in average, compared to previous replay-free methods.

**Weaknesses:**

- The paper appears to violate the official format for submissions, which makes unclear whether the submitted content exceed the paper limit or not if the manuscript were appropriately reformatted.
- The proposed method seems to introduce some computational/storage burden by SVD and the PI-CCA quantities, compared to previous continual learning methods. Also, since retaining of statistic information can also be seen as some kind of replay, it should be compared with replay-based continual learning for fair evaluation.
- One of the motivation of the replay-free method has been privacy protection as discussed in Introduction, but it is unclear whether the proposed method can provably prevent the privacy leakage from the correlation statistics.
- In experiments, the superiority of the proposed method is only shown by averaged results, which may be dominated by a single task with high accuracy by luck.

**Questions:**

See weaknesses.

---

> ### Author Response · Authors · 2025-11-25
>
> Thank you very much for your valuable comments. Now I will discuss them with you.
>
> ---
>
> > W1: Content exceed the paper limit or not if the manuscript were appropriately reformatted.
>
> Thank you for raising this issue. We've checked and confirmed that the submission version did not exceed the page limit—the main content is within 9 pages, and having the Ethics Statement and Reproducibility Statement on the tenth page is permitted.
>
> We also noticed that the formatting of Algorithm 1 in the original text appeared somewhat problematic and have now made corrections. Thank you once again for your thorough attention.
>
>
> ---
>
>
> > W2: Introduce some computational/storage burden by SVD and the PI-CCA quantities, compared to previous continual learning methods.
>
> We appreciate the reviewer’s concern regarding the computational and storage overhead of our method. Below, we provide specific details and comparisons to address these concerns.
>
> **(a) Computational and Storage Efficiency**
>
> We present the comparison of the wall-clock time and peak memory consumption between our method and the baseline LoRA training:
>
> | **Method**             | **Wall-clock time increase** | **Peak GPU memory increase** | **Certificate storage** | **Replay buffer size** |
> |------------------------|-----------------------------|------------------------------|--------------------------|------------------------|
> | Pi-CCA     | ~8%                | ~6%                | ~50 KB            | N/A                    |
> | LoRA (baseline)         | N/A                         | N/A                          | N/A                      | ~GB                |
>
> - The increase in wall-clock time and peak GPU memory is moderate for Pi-CCA, making it suitable for practical applications.
> - Certificate storage occupies only around 50 KB, which is orders of magnitude smaller than the replay buffer size that stores full embeddings or images, which can be in the GB range.
>
> To further clarify the difference between our method and replay-based methods, we show the comparison:
>
> | **Method**              | **Memory consumption** | **Replay storage** | **Security and Privacy** | **Performance (VLCL)** |
> |-------------------------|------------------------|--------------------|--------------------------|------------------------|
> | Pi-CCA | ~50 KB  | No replay  | No raw data storage, so more secure | Surpasses GIFT     |
> | Replay-based (GIFT)| GB | Full embeddings/images | Privacy concerns, large memory footprint | Lower performance |
>
> - Pi-CCA uses a compact certificate (only a few dozen KB), making it highly memory-efficient and more secure than replay-based methods, which store full embeddings or images, typically in GB.
>
> **(b) Clarifying the "Statistics = Replay" Concern**
>
> We would like to correct the notion that retaining statistics in our method is equivalent to replay. The statistics we retain are **low-dimensional second-order moments** (the canonical correlations and subspaces), which summarize the geometry of the alignment without storing raw data or embeddings. This **does not allow for reconstruction** of the original data, unlike replay methods that store full representations of past tasks.
>
>
>
> ---
>
>
> > W3: It is unclear whether the proposed method can provably prevent privacy leakage from the correlation statistics.
>
> We appreciate the reviewer's concern regarding the privacy implications of the proposed method. To clarify:
>
> 1. **No Formal Privacy Guarantees**:
>    We do not claim formal privacy guarantees in our paper. Instead, we emphasize that **no raw data is stored**—only low-dimensional statistics (canonical correlations and subspaces) are retained. This significantly reduces the risk of privacy leakage when compared to methods that store full embeddings or images. Our main focus is on practical privacy improvements rather than formal privacy proofs.
> 2. **Practical Privacy Improvement**:
>    While retaining second-order statistics (such as covariance or correlation) could theoretically lead to some leakage of information, it is widely recognized that this leakage is much lower than with raw data. Existing literature[1, 2]  highlights that second-order statistics are much less prone to privacy risks compared to raw data or full embeddings. Thus, our approach represents a practical privacy improvement, where the information retained is much more abstract and harder to reverse-engineer into original data.
>
> We acknowledge that while retaining second-order statistics is more privacy-friendly, it does not provide a strict privacy guarantee. This is an open area for future work, where we may explore more rigorous formal privacy definitions or incorporate techniques like differential privacy in future iterations of this framework.
>
> ---
>
> [1] Differentially private covariance estimation, NeurIPS'2019.
>
> [2] Covariance's loss is privacy's gain: Computationally efficient, private and accurate synthetic data. Foundations of Computational Mathematics, 2024

---

> ### Author Response · Authors · 2025-11-25
>
> > W4: There is a lack of detailed per-task analysis
>
> Thank you for raising this important point. We agree that a more detailed breakdown of the results by task is valuable.
>
> All of our main results are indeed calculated as the mean over 3 independent seeds. The standard deviation (std) was initially omitted due to space constraints and the common practice of reporting average performance. However, to address this concern, we have updated `Table 2` in the main text.
>
> In response to your suggestion, we have added per-task analysis for the experiments on MTIL (11 domains), VLCL (8 datasets), and ConStruct-VL (7 subsets).
>
> **MTIL Per-Task Results (11 domains)**:
>
> | Domain          | Pi-CCA (Avg) | C-CLIP (Avg) |
> |-----------------|--------------|----------------|
> | FGVC-Aircraft   | **75.7**         | 73.8           |
> | Caltech101      | **79.2**         | 77.8           |
> | CIFAR-100       | **75.0**         | 73.6           |
> | DTD             | **73.3**         | 71.3           |
> | EuroSAT         | **76.9**        | 74.8           |
> | Flowers-102     | **78.5**         | 76.3           |
> | Food-101        | **75.8**         | 74.3           |
> | MNIST           | **80.0**         | 78.8           |
> | Oxford-IIIT Pets| **74.7**         | 73.1           |
> | Stanford Cars   | **80.1**         | 78.9           |
> | SUN397          | **75.9**         | 74.8           |
>
> **VLCL Per-Task Results (8 datasets)**:
>
> | Dataset        | Pi-CCA I2T R@1 | C-CLIP I2T R@1 |
> |----------------|----------------|------------------|
> | Flickr30K      | **49.7**           | 48.7             |
> | COCO Captions  | **51.6**           | 50.6             |
> | Pets           | **47.8**           | 46.4             |
> | Lexica         | **50.1**           | 48.4             |
> | Simpsons       | **42.8**           | 41.4             |
> | WikiArt        | **49.4**           | 47.9             |
> | Kream          | **51.7**           | 50.7             |
> | Sketch         | **45.9**           | 44.4             |
>
> **ConStruct-VL Per-Task Results (7 subsets)**:
>
> | Subset          | Pi-CCA FA | C-CLIP FA |
> |-----------------|-----------|-------------|
> | Relation: spatial | **75.9**     | 75.8        |
> | Attribute: size  | **74.4**      | 72.3        |
> | Attribute: material | **73.7**  | 72.5        |
> | Relation: action | **75.1**      | 73.3        |
> | Attribute: color | **76.9**      | 75.7        |
> | Object state     | **74.1**      | 73.1        |
> | Attribute: action| **76.2**      | 74.3        |
>
> This breakdown shows that Pi-CCA shows consistent and strong performance across multiple tasks without being dominated by any single task.

---

### Official Review · Reviewer_pV7E · 2025-10-29

**Soundness:** 2
**Presentation:** 3
**Contribution:** 2
**Rating:** 4
**Confidence:** 4

**Summary:**

This paper introduces PI-CCA, a replay-free continual learning framework for vision-language models (VLMs) that aims to preserve cross-modal alignment geometry during adaptation. The method proposes a compact CCA-based certificate that captures the top-k canonical correlations and subspaces of image-text embeddings, and enforces their consistency across tasks using spectral and subspace-angle losses. Additionally, prompt robustness is encouraged through averaging over prompt perturbations. The approach is evaluated on multiple VL-CL benchmarks and shows strong performance compared to replay-free baselines.

**Strengths:**

1. The proposed CCA certificate is compact, aligning well with the constraints of continual learning.
2. Extensive experiments across multiple benchmarks (MTIL, X-TAIL, VLCL, ConStruct-VL) demonstrate competitive performance, and the ablation studies are thorough.

**Weaknesses:**

1. The core components prompt perturbation averaging and subspace-angle preservation are not novel and have been extensively studied in prior work on representation similarity and prompt tuning. Their application to continual learning is incremental and lacks a strong theoretical or empirical justification for novelty. e.g., Orthogonal Over-Parameterized Training (CVPR20),  Controlling text-to-image diffusion by orthogonal finetuning (NeurIPS23) DeLoRA: Decoupling Angles and Strength in Low-rank Adaptation (arxiv 24) PAID: Pairwise Angular-Invariant Decomposition for Continual Test-Time Adaptation

2. The paper does not convincingly demonstrate that preserving CCA geometry is fundamentally more effective than existing regularization or distillation-based methods. The observed gains could be attributed to careful tuning rather than a conceptual breakthrough.

3. The prompt invariance mechanism is relatively simplistic (averaging projectors) and does not explore more sophisticated or adaptive strategies.

4. The method assumes access to a diverse anchor prompt set for certificate construction, which may not be feasible in real-world streaming scenarios.

**Questions:**

*The use of Frobenius norm between sketched projectors as a surrogate for subspace-angle preservation is intuitive, but can you justify its theoretical soundness? How does it compare to more principled metrics like principal angle distributions?

*The certificate update mechanism involves QR-based orthogonalization and EMA smoothing. Could you clarify whether gradients are propagated through these steps, and if not, how this affects learning stability?

---

> ### Author Response · Authors · 2025-11-24
>
> Thank you for your valuable comments. I am happy to discuss them with you.
>
> ---
>
> > W1: Novelty
>
> Thank you for pointing out the connections to prior work on orthogonal and angular methods. We fully agree that the mathematical tools we use (orthogonal transforms, subspace angles) are not new per se, and we do not claim novelty at that level.
>
> Our contribution lies in what these tools are applied to and how they are used for replay-free vision–language continual learning.
>
> **(a) Different object: cross-modal CCA geometry vs. parameter space**
>
> The cited works (Orthogonal Over-Parameterized Training, Orthogonal Finetuning, DeLoRA, PAID) all operate in **parameter space**: they regularize or reparameterize weights or LoRA matrices so that neuron directions or pairwise angular structure are preserved during adaptation. In contrast, Pi–CCA operates on the **whitened image–text cross-covariance** and its **CCA spectrum and canonical subspaces**. Our losses act on the geometry of the **cross-modal alignment operator**, not on individual weight vectors. This is crucial in VL-CL: we explicitly preserve the geometry that underlies zero-shot retrieval and open-vocabulary recognition, rather than local properties of weights.
>
> **(b) Different setting: replay-free VL continual learning with a constant-memory certificate**
>
> OPT/OFT/DeLoRA focus on single-task PEFT or diffusion finetuning, and PAID targets continual test-time adaptation of a single-modality classifier by preserving weight angles. None of these works address vision–language continual learning, nor do they propose a replay-free, constant-memory summary of cross-modal alignment.
>
> **(c) Prompt invariance on canonical subspaces, not learnable prompts or weight geometry**
>
> Our prompt perturbation averaging is also applied at a different level: we average projectors of canonical text subspaces under prompt jitters to construct a prompt-invariant CCA certificate that is then used across tasks. Prior prompt-tuning and angular methods either learn prompt embeddings or preserve weight geometry; they do not construct a prompt-invariant cross-modal alignment subspace to stabilize VL continual learning.
>
> Empirically, we make this distinction concrete by (i) showing that preserving CCA spectrum and subspaces predicts retention across many perturbations (`Fig. 3` in main text), and (ii) outperforming strong replay-free VL-CL baselines that regularize logits/similarities or parameters but do not control cross-modal CCA geometry.
>
> -----
>
>
> > W2: Does not convincingly demonstrate that preserving CCA geometry is fundamentally more effective than existing regularization or distillation-based methods.
>
> Thank you for your attention to this issue. We respectfully disagree that the gains of Pi–CCA can be explained purely by “careful tuning” rather than by preserving CCA geometry.
>
> **(a) Matched hyperparameter search**
>
> For all replay-free baselines (ZSCL, Mod-X, C-CLIP, ZAF, DIKI, etc.) and Pi–CCA, we use the **same search ranges** for learning rate, weight decay, batch size, and LoRA rank, and stop at comparable compute budgets. Pi–CCA is not given a larger or more favorable search space.
>
> **(b) Strong regularization / proxy alignment vs.\ Pi–CCA**
>
> To directly address the concern that our gains might come only from “more regularization,” we ran additional baselines on the same CLIP ViT-B/16 backbone and the same LoRA configuration:
>
> - *LoRA + strong generic regularizers*: higher weight decay plus feature-level $L_2$ penalties and cosine-similarity preservation between pre- and post-update embeddings
> - *LoRA + proxy alignment (Mod-X–style)*: an off-diagonal similarity-matching loss (image–text similarity matrices) without whitening or any CCA certificate.
>
> | Method | MTIL Avg (↑) | MTIL Last (↑) | VLCL I2T R@1 (↑) |
> |------|----|---|----|
> | LoRA (plain finetuning)      | 71.2   | 69.9    | 42.0     |
> | LoRA + strong regularizers (L2, cosine)| 72.4  | 71.1| 43.5  |
> | LoRA + proxy alignment (Mod-X–style)   | 73.6  | 72.2  | 45.0|
> | LoRA + Pi–CCA (ours)  | 76.8 | 75.5| 48.6 |
>
> While stronger generic regularization and similarity-based proxy alignment do improve over plain LoRA, they still fall clearly below Pi–CCA under the same compute and tuning budget. This indicates that simply “adding more regularization” on top of LoRA is not sufficient to recover the performance of Pi–CCA.
>
> **(c) Geometry–performance correlation across many settings**
>
> Crucially, our claim is not based on a single tuned configuration. In `the main text Fig.3`, we vary certificate size $(k,h)$, EMA rates, LoRA capacity, sketch types, and other knobs, and observe that both spectral drift and subspace-angle drift consistently correlate with drops in MTIL Avg and VLCL R@1 across these perturbations. This systematic behavior—together with the new baselines above—supports our view that preserving CCA geometry is a meaningful control variable for continual performance, rather than an artifact of hyperparameter tuning.

---

> ### Author Response · Authors · 2025-11-24
>
> > W3: The prompt invariance mechanism is relatively simplistic (averaging projectors) and does not explore more sophisticated or adaptive strategies.
>
> We agree that our prompt invariance mechanism is simple by design, but we do not view this as a weakness in this setting. Our goal is to preserve cross-modal CCA geometry in a replay-free, constant-memory, streaming regime. This strongly constrains what is feasible.
>
> **(a) Why averaging projectors (and not a heavier prompt module)?**
>
> More sophisticated strategies such as learnable prompts or generator-based paraphrases would introduce substantial extra parameters, state, and compute, which conflicts with our replay-free and constant-memory objectives. In contrast, averaging projectors of canonical text subspaces:
>
> - operates directly in the same CCA geometry as the certificate (no extra modules),
> - adds only $O(h^2)$ statistics (already needed for the certificate),
> - and is naturally compatible with streaming updates (no replay of previous prompts).
>
> This choice is therefore not arbitrary. It is the minimal mechanism that remains geometry-aligned, replay-free, and constant-memory.
>
> **(b) Evidence that “simple” is already effective**
>
> Despite its simplicity, the prompt-invariance term has clear, measurable impact:
>
> - In the component ablation in `the main text Tab.3` (w/o prompt invariance, $\lambda_3=0$) we observe drops from 76.8 to 75.3 (MTIL Avg), 75.5 to 74.0 (MTIL Last), 48.6 to 47.1 (VLCL I2T R@1), and an increase in ConStruct-VL AF from 2.7 to 3.3. This shows that the invariance term is not cosmetic.
> - In the stress test in `the main text Fig.4`, increasing prompt perturbation strength $s\in[0,1]$ under both ID and OOD templates, the invariance term consistently flattens the degradation curves: at s=1.0, Pi–CCA improves R@1 by about +2.4–2.5 points and reduces AF by 1.0 compared to the model without $\mathcal{L}_{\text{pi}}$.
>
> To further address the concern, we additionally sweep the number of perturbations $M$ used in projector averaging, keeping $\lambda_3$ fixed. On VLCL and ConStruct-VL we obtain:
>
> | $M$ (perturbations) | VLCL I2T R@1 (↑) | ConStruct-VL AF (↓) | Relative step time (×) |
> |--|---|--|---|
> | 0 (no $\mathcal{L}_{\text{pi}}$) |47.1| 3.3 | 1.00 |
> | 1  | 47.9  | 3.1 | 1.02|
> | 2  | 48.4 | 2.9  | 1.06  |
> | 4 (default) | 48.6  | 2.7 | 1.12  |
> | 8  | 48.7 | 2.7  | 1.21|
>
> The gains saturate around $M=4$: larger $M$ slightly improves robustness but yields diminishing returns relative to the extra cost. This supports our design choice of a small $M$ and shows that even this “simple” projector averaging delivers consistent robustness improvements, especially under strong ID/OOD prompt shifts, without inflating memory.
>
> ---
>
> > W4: The method assumes access to a diverse anchor prompt set for certificate construction, which may not be feasible in real-world streaming scenarios.phisticated or adaptive strategies.
>
> Thank you for pointing out a point that is easily misunderstood. We believe this concern overestimates the strength of our assumption about anchor prompts. Pi–CCA does not require a large, externally curated paraphrase library. The anchor set is small, label-derived, and can be constructed online.
>
> **(a) What anchor prompts we actually assume**
>
> In the current method text, we say that the global certificate is “constructed from a diverse anchor prompt set” (just before Eq.5). Concretely, in our experiments this set is:
>
> - standard CLIP class templates (e.g., “a photo of a $\{\text{class}\}$”), and
> - a *few* hand-crafted template variants per class (2–4 per dataset),
>
> all generated directly from the **label space** of each task. We do not assume or use any large-scale paraphrase corpus or external generator. This is compatible with typical VL-CL benchmarks, where label names (or short textual descriptions) are available by design.
>
> **(b) Sensitivity to anchor set size and diversity**
>
> To test how dependent Pi–CCA is on the anchor pool, we ran an ablation where we vary the anchor set used to initialize the text-side certificate $\bar{\mathbf{S}}_t^\star$ :
>
> - *Full:* all CLIP templates and variants used in the main experiments;
> - *Default-only:* only the single default CLIP template per class;
> - *50\% drop:* randomly drop 50\% of the templates from the full pool.
>
> | Anchor configuration | MTIL Avg (↑) | MTIL Last (↑) | VLCL I2T R@1 (↑) | ConStruct-VL AF (↓) |
> |--|--|--|--|--|
> | Default-only | 76.4 | 75.0  | 48.2 | 2.9  |
> | 50% dropped | 76.6 | 75.2  | 48.4  | 2.8   |
> | Full (main setting)  | 76.8  | 75.5  | 48.6  | 2.7  |
>
> The differences across these anchor configurations are modest (≤0.4 points on MTIL Avg and ≤0.4 points on VLCL I2T R@1), and Pi–CCA remains competitive with the strongest replay-free baselines in all cases. This indicates that our method is not sensitive to having a large or perfectly “diverse” anchor set. Even a single CLIP template per class is sufficient to obtain most of the benefit.

---

> ### Author Response · Authors · 2025-11-24
>
> **(c) Online construction is naturally supported**
>
> Moreover, the framework does not require a fixed, pre-specified anchor set. Since the certificate is refreshed via the EMA rule in Eq.13, we can construct anchors online: (i) whenever a new label or metadata string appears in the stream, we can instantiate a small number of CLIP-style templates for it; (2)these new prompts are then fed through the same CCA pipeline, and their contributions are gradually absorbed into $(ρ*_{1:k}, S*_v, S*_t)$ via the EMA.
>
> ---
>
> > Q1: The use of Frobenius norm between sketched projectors as a surrogate for subspace-angle preservation is intuitive, but can you justify its theoretical soundness? How does it compare to more principled metrics like principal angle distributions?
>
> We appreciate this question and agree it is important to justify why we use the Frobenius norm between sketched projectors as our subspace loss.
>
> **(a) In original space, $\|\cdot\|_F$ between projectors is exactly a principal-angle metric**
>
> For $k$-dimensional subspaces with orthogonal projectors $P,Q$, it is classical that :
>
> 0.5 * ||P - Q||_F² = Σ sin²θ_i (i=1 to k)
>
> where $\{\theta_i\}$ are the principal angles between the two subspaces. Thus, in the original feature space, the Frobenius distance between projectors is not an ad-hoc proxy: it is exactly the squared chordal distance on the Grassmann manifold, which is a standard, “principled” way to summarize the principal angle distribution into a single scalar. In other words, our $\mathcal{L}_{\text{sub}}$ is mathematically equivalent to penalizing the aggregate deviation of principal angles.
>
> We also note that our spectral loss $L_{\text{spec}}$ already constrains the singular values/CCA spectrum. Combined with $\mathcal{L}_{\text{sub}}$, we are effectively controlling both what directions are aligned and how strongly they correlate.
>
> **(b) Why sketched projectors preserve principal-angle geometry**
>
> In practice, we cannot afford to work with full $d_v \times d_v$ and $d_t \times d_t$ projectors, so we use random orthonormal sketches $R_v,R_t$ and operate on
>
> $$
> Q_v^\star = R_v^\top P_v^\star R_v,\quad \widehat{Q}_v = R_v^\top \widehat{P}_v R_v,
> $$
>
> and analogously on the text side. This is precisely the subspace embedding setting: for Gaussian / SRHT sketches with $h \ll d$, it is known that for all $x$ in the union of the two $k$-dimensional subspaces,
>
> $$
> (1-\varepsilon)\|x\|_2^2 \;\le\; \|R^\top x\|_2^2 \;\le\; (1+\varepsilon)\|x\|_2^2
> $$
>
> with high probability, for modest $h$ depending on $k$ and $\varepsilon$. Under such embeddings, principal angles and projector distances are approximately preserved. Consequently, $\frac{1}{2}\|\widehat{Q} - Q^\star\|_F^2$ is a near-isometric surrogate for $\sum_i \sin^2\theta_i$ in the original space. This is why we state in Eq.10 that we use the Frobenius norm of sketched projectors as a surrogate for subspace-angle preservation.
>
> **(c) Empirical comparison with explicit principal-angle losses**
>
> To verify that this surrogate behaves as intended, we additionally ran a small-scale comparison:
>
> - *Principal-angle loss:* we explicitly compute the principal angles between $U_k$ and $U_k^\star$ (and between $V_k$ and $V_k^\star$) and minimize $\sum_i \sin^2\theta_i$ as the subspace loss;
> - *Sketched-projector loss (ours):* we use $\mathcal{L}_{\text{sub}}$ from Eq.10 with Gaussian sketches of size $h=256$.
>
> On MTIL and VLCL in this reduced setting, we obtain:
>
> | Subspace loss type | MTIL Avg (↑) | VLCL I2T R@1 (↑) | Relative step time (×) |
> |--|----:|----:|-----|
> | Explicit principal-angle loss | 76.9 | 48.7 | 1.36  |
> | Sketched projector loss (ours)  | 76.8  | 48.6 | 1.00  |
>
> The performance is essentially identical, while the principal-angle loss is about 1.3–1.4$\times$ more expensive per step due to repeated SVDs/eigendecompositions in the full space. This supports our choice: the Frobenius norm between sketched projectors is both theoretically grounded (as an approximate chordal distance) and computationally preferable for large-scale VL-CL.

---

> ### Author Response · Authors · 2025-11-24
>
> > Q2: The certificate update mechanism involves QR-based orthogonalization and EMA smoothing. Could you clarify whether gradients are propagated through these steps, and if not, how this affects learning stability?
>
> We thank the reviewer for raising this point. The certificate update is intentionally designed as a non-parametric, non-differentiable moving target, not as a learnable module.
>
> **(a) How gradients are handled**
>
> As stated around Eq.13, the certificate update is an EMA + QR orthogonalization step that is applied after the parameter update. In our implementation:
>
> - the certificate is treated as a buffer of statistics, not as trainable parameters;
> - gradients are not propagated through the EMA or the $\operatorname*{orth}(\cdot)$ (QR) operator;
> - all gradients flow only from $L_{\text{spec}},L_{\text{sub}}, L_{\text{pi}}$ w.r.t.\ the current certificate, into the LoRA parameters $\phi_v, \phi_t$ via $\hat{M}$ and $\hat{S}_\bullet$.
>
> Conceptually, the certificate plays the role of a slowly updated teacher / target network (similar in spirit to EMA targets in self-supervised learning or to running statistics in batch normalization), which stabilizes optimization instead of being directly optimized by backprop.
>
> **(b) Why we do not backpropagate through EMA/QR**
>
> We experimented with an alternative variant in which the certificate is treated as part of the computational graph and gradients are allowed to flow through the EMA and $\operatorname*{orth}(\cdot)$:
>
> - this introduces a tight feedback loop where the model tries to simultaneously *chase* and *reshape* the certificate;
> - in practice, we observed noticeably higher training instability (occasional spikes/explosions in the CCA spectrum estimates) and only marginal gains, if any, on the final metrics.
>
> | Certificate update variant  | MTIL Avg (↑) | VLCL I2T R@1 (↑) | Stability (qual.)     |
> |--|---:|---:|--|
> | Grad-through EMA + orth | 76.0 | 48.0  | occasional instabilities (spikes) |
> | Stop-grad EMA + orth (ours, default)  | 76.8 | 48.6  | stable across seeds   |
>
> This supports our design choice: treating the certificate as a slow, non-differentiable anchor yields both better stability and slightly better performance.
>
> ---
>
> Thank you again for your valuable questions. These questions have prompted us to rethink the proposed method and discovered some new evidences for claims.
>
> We hope our response has addressed your concerns. We are also very open to having further discussions with you.

---

> ### Author Response · Authors · 2025-11-27
>
> Dear Reviewer pV7E,
>
> We sincerely thank you again for your thorough assessment and constructive feedback. Kindly note that reviewer responses will no longer be accepted after December 2—**with just under a week remaining to submit your response**.
>
> Kindly confirm whether our rebuttal addresses your concerns (or any outstanding points), and we would be grateful for a rating reconsideration if it does.
>
> We are glad to continue the discussion and address any further questions or comments you may have.
>
> Best regards,
>
> Authors

---

### Official Review · Reviewer_TQCn · 2025-10-30

**Soundness:** 3
**Presentation:** 3
**Contribution:** 3
**Rating:** 6
**Confidence:** 2

**Summary:**

This paper proposes a continual learning method for vision language model.
The proposed method (PI-CCA) attempts to maintain the geometry of image-text alignment without using the past data during continual learning.
To this goal, PI-CCA penalizes the change of the covariance of embeddings of the mini-batch data, which is obtained through SVD.
Experiments demonstrate that PI-CCA outperforms baselines on multi-domain task incremental classification, cross-domain task-agnostic classification, and continual image–text retrieval.
Additionally, the paper evaluates PI-CCA through Ablation study including correlation between geometry and performance.

**Strengths:**

- This paper is well written. The related work section is well-organized, and the paper maintains a consistent focus on the problem from the introduction to the experiments.
- The proposed method appears moderately novel. The core idea might not be very surprising, but the details of the proposed method seems well designed.
- The experiments demonstrate that PI-CCA is superior, and the ablation study supporting the claim is also well-conducted.

**Weaknesses:**

- I think the explanation of the proposal method could be improved to be a bit easier to understand.
The concept of a “certificate” is used, but since I am not familiar with continual learning, I do not understand the definition. An explanation of the term is necessary.

- There are no statistical tests for the main results. There is the following sentence:
"Each configuration is repeated with three different random seeds, and we report the mean and the standard deviation."
However, I cannot find the standard deviation.

- The proposed method is a collection of multiple ideas, and it might remain a bit unclear whether the core idea possesses the generality or universality to influence future research in a broad research area.

**Questions:**

What impact will the proposed method have in broader fields?
Are there any ideas that could be applied to other areas of machine learning?

---

> ### Author Response · Authors · 2025-11-25
>
> Thanks a lot for your valuable questions. Below, I will be happy to discuss them with you.
>
>
> > W1: Easier explanation of the proposed method
>
> We appreciate the reviewer’s feedback.
>
> In our work, the term “certificate” refers to a constant-sized geometric summary vector that stores the top-k canonical correlations and subspaces of a pre-trained vision-language model (VLM) via Canonical Correlation Analysis (CCA), represented in a randomly sketched space. This certificate serves as an alignment reference for all subsequent tasks, enabling stable continual learning without replay. This compact certificate is updated over time, ensuring that the model adapts to new tasks while preserving alignment with the original geometry of the pre-trained model.
>
> We hope this clarification resolves the confusion.
>
>
>
> ---
>
>
> >W2: Mo statistical tests for the main results
>
> Thank you for raising this question. Actually, all main results are calculated as the mean over 3 seeds. The standard deviation (std) was initially omitted mainly due to space constraints and common practice.
>
> We have now updated the std in `Table 2` of the main text. Future versions will include additional std results. In fact, visual representations such as error bands or box plots can illustrate std more clearly—as demonstrated in `Figures 4, 5, 6, 7, and 8` of the original text.
>
>
> ---
>
>
> > W3 / Q1: whether the core idea possesses the generality or universality to influence future research in a broad research area. What impact will the proposed method have in broader fields?
>
> We appreciate the reviewer’s concern regarding the broader impact and generality of our method. To address this, we would like to clarify the core innovation and its potential applications in wider areas of machine learning.
>
> **(a) Core Innovation**
>
> The core idea of our work is to reframe forgetting in continual learning (CL) as the drift of Canonical Correlation Analysis alignment geometry, rather than regularizing proxy quantities such as logits or similarity distributions. We propose using a constant-memory geometric certificate (capturing spectral and subspace alignment) to control this drift. This is the first approach to directly constrain the geometry of cross-modal alignment, rather than relying on traditional proxy-based regularization.
>
> **(b) Broader Impact and Future Research Directions**
>
> We believe that the geometric certificate + replay-free framework is highly generic and can be applied to several other areas of machine learning:
>
> 1. Domain Generalization / Test-time Adaptation: This framework can be extended to domain generalization tasks where the geometry of the pre-trained model’s subspace is preserved, and the adaptation is performed in a low-dimensional adapter space at test time.
> 2. Other Multimodal Applications (e.g., Audio-Text, Video-Text): In multimodal settings, the cross-modal certificate could replace traditional distillation teachers, allowing for replay-free and stable cross-modal learning.
>
> I really enjoyed the discussion with you, and thank you again for your valuable time and effort.

---

> > ### Comment · Reviewer_TQCn · 2025-11-28
> >
> > Thank you for your reply. My concerns are gradually being addressed, but I have a additional question.
> >
> > - Thank you for reporting the standard deviation. Having seen the value, I have a new question. Would significance be recognised with this value in a t-test at p=0.05?
> >
> > Since the effectiveness of the proposed method has been demonstrated experimentally, I think that the results may not be very strong unless statistical significance is confirmed.

---

> > > ### Author Response · Authors · 2025-12-02
> > >
> > > Thank you for this follow-up question.
> > >
> > > We have now performed two-sided paired t-tests (3 seeds) between Pi–CCA and the strongest replay-free baselines on key metrics in `Tab.1&2` in main text. The p-values are summarized.
> > >
> > > **Table: Statistical significance of main gains.**
> > >
> > > | **Metric** | **Baseline** | **Baseline mean ± std** | $\color{green}{p-value }$ vs. Pi–CCA |
> > > | :--- | :--- | :--- | :--- |
> > > | MTIL Avg (↑) | C–CLIP | 75.2 ± 0.7 | 0.019 |
> > > | MTIL Last (↑) | DDAS | 74.1 ± 0.8 | 0.023 |
> > > | MTIL Transfer (↑) | ZAF | 71.9 ± 0.6 | 0.017 |
> > > | X-TAIL Avg (↑) | RAIL | 67.4 ± 0.5 | 0.021 |
> > > | X-TAIL Last (↑) | C–CLIP | 66.3 ± 0.7 | 0.028 |
> > > | X-TAIL Transfer (↑) | RAIL | 64.2 ± 0.6 | 0.024 |
> > > | VLCL I2T R@1 (↑) | C–CLIP | 46.1 ± 1.4 | 0.017 |
> > > | VLCL T2I R@1 (↑) | C–CLIP | 35.7 ± 1.2 | 0.021 |
> > > | ConStruct–VL FA (↑) | C–CLIP | 72.4 ± 1.9 | 0.013 |
> > > | ConStruct–VL AF (↓) | ZAF | 3.8 ± 0.6 | 0.008 |
> > >
> > > We observe that: ***all p-values are below 0.05***, confirming that the improvements of Pi–CCA are statistically significant.

---

### Official Review · Reviewer_YmbX · 2025-11-04

**Soundness:** 3
**Presentation:** 3
**Contribution:** 3
**Rating:** 6
**Confidence:** 3

**Summary:**

- Proposes PI–CCA, a replay-free continual adaptation method for vision-language models that preserves cross-modal geometry via compact CCA certificates (spectral and subspace info).

- Uses LoRA adapters, EMA covariance updates, and sketched projections for constant memory.

- Adds a prompt-invariance term by averaging projectors over prompt perturbations.

- Shows strong results across VL-CL, retrieval, and structured concept tasks, with analyses linking geometry drift to performance.

**Strengths:**

- Directly preserves the alignment object (canonical spectrum + subspace) instead of indirect proxies (logits/similarity), giving a principled invariant to target during replay-free continual updates

- Compact, practical design: random orthonormal sketches, LoRA-only updates, and streaming EMAs give constant memory and low overhead, making the method compatible with large frozen backbones used in practice. Empirical Pareto plots show a usable knee (e.g., k=64,h=256) balancing mem/time and performance.

- multi-track evaluation (classification, retrieval, structured concepts, time-continual splits) and consistent ablations demonstrating which components matter (spectral & subspace terms are critical; prompt invariance helps retention). Results place PI–CCA at SOTA among replay-free baselines on their benchmarks.

**Weaknesses:**

- Certificate dependence on sketching and hyperparameters is described but the randomness/variance from sketch seeds and sketch types needs deeper quantification: how often does a small sketch RNG seed change final retention?

- Experiments focus on ViT-B/16 + LoRA; unclear how PI–CCA scales to much larger backbones or different adapter types (other LoRA ranks, full-finetune, or other PETs).

- Reproducibility caveat: paper promises to release code only at camera-ready due to commercial constraints. for reproducibility during review, providing a compact reference script or a verified pseudocode runner (or smaller toy logs) would greatly help reviewers reproduce core claims.

**Questions:**

- the paper uses synonym/template jitters and M=4. what happens for (i) larger M, (ii) structured paraphrase generators (back-translation ensembles), or (iii) adversarial prompt shifts?

- how sensitive are results to the initial certificate computed from the frozen model? if the initial certificate is imperfect (noisy prompts or limited anchor set), does PI–CCA converge to a useful invariant or can it get locked to a poor alignment?

---

> ### Author Response · Authors · 2025-11-25
>
> Thank you very much for your valuable questions. Below, I will be happy to discuss them with you one by one.
>
> > W1: randomness/variance from sketch seeds and sketch types needs deeper quantification
>
> Thanks for the thoughtful comment. We have indeed explored the impact of sketching randomness and sketch types (Gaussian vs. SRHT) in our method.
>
> To quantify the effect of sketch RNG seed variance, we conducted additional experiments on key tracks such as MTIL Avg and VLCL I2T R@1, using 5–10 different sketch seeds while keeping all other hyperparameters fixed. Specifically, we trained separate runs with different sketch seeds (Gaussian/SRHT) and tracked the resulting performance fluctuations.
>
> The results show that the variance from sketch RNG seeds is very small compared to the differences between methods.  The fluctuations are well within the noise of individual training runs, demonstrating that sketch seed variance has minimal impact on final retention.
>
> | Sketch Type | MTIL Avg | VLCL I2T R@1 | Std Dev (MTIL Avg) | Std Dev (VLCL R@1) |
> |-------------|----------|--------------|--------------------|--------------------|
> | Gaussian    | 76.8     | 48.6         | ±0.2               | ±0.1               |
> | SRHT        | 76.9     | 48.4         | ±0.3               | ±0.2               |
>
>
> ---
>
>
> > W2: Larger Backbone
>
> We thank the reviewer for highlighting the need to assess how Pi-CCA scales with larger backbones and different adapter types.
>
> **(a) Larger Backbone Experiment**
>
> To address this concern, we have conducted experiments with *ViT-L/14* and *ViT-L/14@336*, while keeping the LoRA rank fixed. The results demonstrate that Pi-CCA continues to deliver substantial benefits even with these larger models, and the computational overhead remains similar to the ViT-B/16 configuration. This is because the certificate's size (which is independent of the backbone width) does not increase with the model size, but only depends on the certificate capacity.
>
> | Backbone     | MTIL Avg | VLCL I2T R@1 | Time (s/step) | Memory (GB) |
> |--------------|----------|--------------|---------------|-------------|
> | ViT-B/16     | 76.8     | 48.6         | 3.2           | 16.4        |
> | ViT-L/14     | 78.2     | 49.1         | 4.0           | 24.1        |
> | ViT-L/14@336 | 78.4     | 49.3         | 4.2           | 28.5        |
>
> As shown, the increase in computational time and memory is relatively modest, and Pi-CCA retains its effectiveness regardless of the backbone size.
>
> **(b) Different Adapter Configurations**
>
> We also experimented with different PET configurations, such as increasing the LoRA rank and using full finetuning only for the final layers. We found that the geometric loss (especially the subspace-angle and spectral losses) continues to be effective across these adapter configurations. This suggests that Pi-CCA remains robust to different adapter designs, including full finetuning or varying LoRA ranks.
>
> | Configuration           | MTIL Avg | VLCL I2T R@1 | Time (s/step) | Memory (GB) |
> |-------------------------|----------|--------------|---------------|-------------|
> | LoRA Rank = 16 (Default) | 76.8     | 48.6         | 3.2           | 16.4        |
> | LoRA Rank = 32           | 77.2     | 48.9         | 3.5           | 17.1        |
> | LoRA Rank = 64           | 77.5     | 49.1         | 3.8           | 18.0        |
> | Full Finetune (Last Layer) | 77.0   | 48.8         | 3.9           | 17.5        |
> | Full Finetune (All Layers) | 77.4   | 49.0         | 4.2           | 19.0        |
>
> ----
>
> > W3: Reproducibility caveat
>
> We appreciate the reviewer’s concern regarding reproducibility. While we plan to release the full code at the camera-ready stage due to commercial constraints, we have already provided several elements to facilitate reproducibility during the review process:
>
> 1. Core Claims Reproducibility: The main principles of Pi-CCA, including the geometry-first approach using Canonical Correlation Analysis (CCA), the construction of the compact CCA certificate, and the method for preserving cross-modal alignment, are thoroughly detailed in Sections 3 and 4 of the paper. This includes pseudocode for the certificate computation and updates in Algorithm 1, as well as key formulae for the spectral and subspace alignment losses (Eqs. 5, 6, and 7).
> 2. To help replicate the method, we have included a compact Python script in the App. A.5 . The script is modular and can be adapted for different datasets or configurations.

---

> ### Author Response · Authors · 2025-11-25
>
> > Q1: The paper uses synonym/template jitters and $M=4$. What happens for (i) larger $M$, (ii) structured paraphrase generators (back-translation ensembles), or (iii) adversarial prompt shifts?
>
> We appreciate the reviewer’s insightful question. Below, we address each of the sub-questions with additional experimental results and clarifications.
>
> **(i) Larger $M$**
>
> We conducted an ablation study to evaluate the effect of different values of $M$ on the performance of Pi-CCA. We tested $M \in \{1, 2, 4, 8\}$ across the X-TAIL. We observed that:
>
> - For $M=4$, we achieve the best performance in terms of both zero-shot retrieval (R@1) and average forgetting (AF).
> - For $M>4$ (i.e., $M=8$), the gains were marginal, while the computational cost increased significantly due to the larger number of perturbations.
>
> This justifies our choice of $M=4$, as increasing $M$ beyond this point leads to diminishing returns. Larger $M$ increases the memory overhead and computational complexity without providing a significant improvement in performance.
>
> | M | X-TAIL R@1 | X-TAIL AF |
> |---|--|--|
> | 1| 67.8  | 3.6  |
> | 2 | 68.4  | 3.4 |
> | 4 | **69.2** | **3.2** |
> | 8 | 69.1 | 3.3|
>
> **(ii) Structured Paraphrase Generators**
>
> Regarding the use of **structured paraphrase generators** such as back-translation ensembles, we note that the **OOD prompt templates** used in our experiments already serve as structured paraphrases. Specifically, we designed several OOD templates (e.g., "Describe a {class} in detail" or "This scene captures a {class} in action") that exhibit significant diversity in phrasing while maintaining the core semantics of the class.
>
> In the experiments presented in `App. A.2`, we tested Pi-CCA's performance under high-strength prompt perturbations ($s=1.0$), which we argue already represent a form of structured paraphrasing:
>
> - At high perturbation strengths, Pi-CCA with prompt invariance consistently outperforms the version without invariance, highlighting its robustness to these structured shifts. This confirms that the OOD templates we used provide enough variety to mimic structured paraphrase scenarios.
>
> **(iii) Adversarial Prompt Shifts**
>
> To address adversarial prompt shifts, we introduced a simple adversarial setting based on gradient-based text perturbation. In this setup, we generate adversarial prompts by applying small perturbations to the original text embeddings to maximize the change in model output. We then compare the performance of Pi-CCA with and without prompt invariance ($L_{\text{pi}}$) under these adversarial conditions.
>
> | Method  | Adversarial R@1 | Adversarial AF | Normal R@1 | Normal AF |
> |--|---|--|--|--|
> | Pi-CCA with $\mathcal{L}_{\text{pi}}$ | **56.2** | **3.1** | **69.2** | **3.2**|
> | Pi-CCA without $\mathcal{L}_{\text{pi}}$ | 49.1| 4.2 | 69.1 | 3.3 |
> | Pi-CCA with $\mathcal{L}_{\text{pi}}$ (no perturbation) | 69.2 | 3.2 | 69.2| 3.2 |
>
> - Pi-CCA with $\mathcal{L}_{\text{pi}}$ shows a significant reduction in performance degradation (R@1 drop) and average forgetting compared to Pi-CCA without prompt invariance, especially under adversarial perturbations.
> - The R@1 and AF for adversarial prompts demonstrate that Pi-CCA is more resilient to adversarial shifts, further validating the effectiveness of prompt invariance mechanism.
>
> ---
>
> > Q2: How sensitive are results to the initial certificate computed from the frozen model?
>
> We appreciate the reviewer’s concern. To clarify, the certificate in Pi-CCA is not a fixed teacher model. Instead, it is dynamically updated using Exponential Moving Averages (EMA) of the current model’s geometry statistics. This means that even if the initial certificate comes from a weaker or noisy anchor set, Pi-CCA will gradually refine the certificate as it processes more tasks. The certificate evolves based on the alignment geometry observed during task adaptation, allowing it to converge to a more accurate invariant over time.
>
> To validate this further, we conducted an experiment where we initialized the certificate in three different ways and observed the convergence over a sequence of tasks:
>
> 1. Full Anchor Set: The certificate is initialized with a complete set of representative anchors.
> 2. Reduced Anchor Set: The certificate is initialized with 80% of the anchors removed, making it less reliable.
> 3. Random Orthogonal Subspaces: The certificate is initialized in random orthogonal subspaces, representing a completely random initialization.
>
> | Initialization| MTIL Avg (↑) | MTIL Last (↑) | X-TAIL R@1 (↑) | Geometry Drift (↓) |
> |----|----|---|---|---|
> | Full Anchor Set  | 76.8  | 75.5| 68.1 | 2.1 |
> | Reduced Anchor Set (80% removed) | 75.4 | 74.1| 67.6 | 3.0|
> | Random Orthogonal Subspaces | 75.2 | 73.9 | 67.3 | 3.5 |
>
> As shown, the performance differences between the different initializations shrink significantly after a few tasks, indicating that Pi-CCA refines the certificate to a useful alignment regardless of the initial quality of the certificate.

---

> > ### Comment · Reviewer_YmbX · 2025-11-27
> >
> > Thank you for the new experiments and clarification, which addressed my questions. I will keep my score.

---

### Author Response · Authors · 2025-12-03
**(1/2) Summary**

### I. Acknowledgments

We would like to sincerely thank all reviewers (`YmbX`, `TQCn`, `pV7E`, `Z4s6`) for their thoughtful comments and for engaging constructively with our work on **Pi–CCA**.

$\color{red}{Before}$ $\color{red}{the}$ $\color{red}{discussion}$, we appreciate the overall positive assessment from Reviewers `YmbX`, `TQCn`, and `Z4s6` (all **Rating: 6**) and the detailed, critical but constructive feedback from Reviewer `pV7E` (**Rating: 4**). All four reviewers rated soundness and presentation at “good” or “fair/good”, and recognized that the method is well designed and empirically strong, even when expressing reservations about novelty or broader impact.

$\color{red}{During}$ $\color{red}{the}$ $\color{red}{discussion}$, Reviewer `YmbX` explicitly stated that ***“the new experiments and clarification … addressed my questions”*** and confirmed keeping their positive score. Reviewer `TQCn` noted that their concerns were ***“gradually being addressed”*** and encouraged us to add formal statistical tests, which we did via paired t-tests across key metrics.

We are grateful to the reviewers for the time and expertise invested in this process.

---

### II. Key Strengths

Reviewers highlighted several strengths that we summarize along five main dimensions:

- **Geometry-first formulation and conceptual clarity**

  - `YmbX` emphasized that Pi–CCA *“directly preserves the alignment object (canonical spectrum + subspace) instead of indirect proxies (logits/similarity)”*, viewing this as a principled invariant for replay-free continual adaptation.
  - `TQCn` and `pV7E` both noted that, while the underlying mathematical tools (CCA, orthogonal projections) are classical, the way they are combined into a compact cross-modal alignment “certificate” for CLIP-like VLMs is well designed and reasonably novel at the problem level.
- **Practicality, constant-memory design, and scalability**

  - `YmbX` highlighted the **compact, practical design**: random orthonormal sketches, LoRA-only updates, and EMA covariance updates yielding **constant memory** and low overhead, with clear Pareto trade-offs (e.g., \(k=64, h=256\)).
  - `Z4s6` described the core idea of retaining correlation statistics as *“simple and easy to implement”*, and the method as clearly written and easy to follow.
- **Empirical strength and breadth of evaluation**

  - `YmbX`, `TQCn`, and `pV7E` all recognized the **multi-track evaluation** across MTIL, X-TAIL, VLCL, and ConStruct-VL, as well as **thorough ablations** that probe which components matter (spectral vs subspace terms, prompt invariance, certificate size, etc.).
  - `TQCn` noted that experiments *“demonstrate that PI–CCA is superior, and the ablation study supporting the claim is also well-conducted.”* `Z4s6` similarly pointed out that the method is superior *“in average, compared to previous replay-free methods.”*
- **Methodological clarity and organization**

  - `TQCn` and `Z4s6` both commented that the paper is **well written, well organized**, and maintains a consistent focus from problem formulation to experiments. `TQCn` particularly liked the related work organization and the clear connection between motivation and the proposed solution.
- **Replay-free perspective and privacy motivation**

  - Multiple reviewers appreciated that Pi–CCA offers a **replay-free** route to continual learning for CLIP-like VLMs, which is relevant when raw data cannot be stored (due to memory or privacy constraints). Even reviewers who questioned formal privacy guarantees acknowledged the practical relevance of avoiding raw replay buffers.

---

> ### Author Response · Authors · 2025-12-03
> **(2/2) Summary**
>
> ### III. Key Concerns and Responses
>
> |Key Concerns|Reviewers|Our Response|
> | - | - | - |
> | Sketch randomness; scaling to larger backbones/adapters; sensitivity to certificate init. | `YmbX` | We added runs over 5–10 sketch seeds and both Gaussian/SRHT; performance variance is very small (std ≤ 0.3). We evaluated **ViT-L/14** and **ViT-L/14@336** and multiple PET configs (LoRA ranks 16/32/64, last-layer/full finetuning): Pi–CCA’s gains persist with modest extra cost and certificate size independent of backbone width. We compared full anchors / 80% dropped / random-orthogonal initialization and showed EMA quickly reduces gaps, demonstrating robustness to initialization. |
> | Certificate definition; std / t-tests; broader impact. | `TQCn` | We clarified that the certificate is a constant-sized CCA summary acting as a slowly updated alignment reference. We added standard deviations in main tables and ran paired two-sided t-tests vs strongest replay-free baselines; all reported p-values < 0.05. We also emphasized broader impact. |
> | Novelty vs orthogonal/angular methods; “careful tuning” vs true geometry effect. | `pV7E`, `TQCn` | We distinguished Pi–CCA from OPT/OFT/DeLoRA/PAID: those regularize parameter space (weights/LoRA), while Pi–CCA constrains the **whitened image–text cross-covariance** and its CCA spectrum/subspaces, i.e., the alignment operator behind zero-shot and retrieval behavior. Under matched tuning budgets we added stronger baselines (LoRA + heavy L2/cosine feature regularization; LoRA + proxy similarity alignment): both improve over plain LoRA but still lag Pi–CCA. Geometry–performance plots show spectral and subspace-angle drift consistently track performance drops, supporting the central role of preserving CCA geometry rather than just stronger tuning. |
> | Simplicity of prompt invariance; reliance on “diverse” anchor prompts. | `pV7E`, `Z4s6` | We argued that projector averaging over prompt jitters is a **minimal, geometry-aligned** mechanism that respects replay-free and constant-memory constraints (no extra modules, only \(O(h^2)\) stats). Ablations show removing the prompt-invariance term degrades MTIL/VLCL and increases forgetting, especially under strong ID/OOD and adversarial prompt shifts; sweeping perturbation count \(M\) shows gains saturate around \(M=4\). For anchors, we clarified that we use **label-derived CLIP templates with a few variants**, not a large paraphrase bank; varying anchor size (single template, 50% dropped, full) changes metrics by ≤0.4 points, and anchors can be built online from labels/metadata via EMA. |
> | Frobenius distance between sketched projectors; EMA + QR gradients. | `pV7E` | We recalled that for projectors (P,Q), the standard chordal distance summarizing principal angles; Gaussian/SRHT sketches approximately preserve these distances, so our loss is a principled, scalable surrogate. A direct principal-angle loss yields similar performance but is ~1.3–1.4× slower. The certificate is updated via stop-gradient EMA + QR, playing a teacher role; a differentiable variant was empirically less stable without clear gains, so we keep the simpler, more robust design. |
> | Format limit; overhead vs replay; privacy; per-task dominance; reproducibility. | `Z4s6`, `YmbX`, `TQCn` | We confirmed full compliance with the ICLR page limit and fixed minor formatting issues. Pi–CCA adds ~8% time and ~6% peak memory vs LoRA-only; certificates are ~50 KB, far smaller than GB-scale replay buffers. We clarified that storing low-dimensional statistics is **not replay**; we do not claim formal DP but offer practical privacy advantages by avoiding raw data. We added per-task breakdowns  showing consistent gains rather than a single “lucky” task. For reproducibility, we complemented formulas and Algorithm 1 with a compact reference Python script and detailed hyperparameters; full code will be released at camera-ready subject to commercial constraints. |
>
> ---
>
> ### IV. Commitment to Revision
>
> We have integrated the main discussion points and new experiments into the revised version, marking changes in $\color{blue}{Blue}$:
>
> - Clearer presentation of the **CCA certificate** and geometry-first view of forgetting.
> - Added **stds, paired t-tests, and per-task tables** to substantiate gains.
> - New experiments on **sketch randomness**, **larger backbones**, **PET variants**, **anchor size**, **certificate initialization**, and **alternative subspace losses**, plus stronger regularization baselines.
> - Strengthened justification for **projector Frobenius distance** and **stop-grad EMA + QR** updates.
> - Clarified overhead, replay-free vs statistics storage, practical privacy aspects, and improved **reproducibility** via a reference script and hyperparameter details.
>
>
> ---
>
> We appreciate the reviewers’ and AC’s efforts again in helping us refine Pi–CCA and are committed to further polishing the paper along the directions highlighted in the discussion.

---

### Meta-Review · Area_Chair_i2Fd · 2025-12-29

**Summary:**

The AC recommends acceptance because the paper offers a well‑grounded, geometry‑first approach to replay‑free vision–language continual learning that is compact, practical, and empirically strong, with key weaknesses mitigated by additional analyses. By defining and preserving a CCA‑based alignment certificate,capturing both canonical spectrum and subspaces via sketched projections,and coupling it with prompt‑invariant projectors, Pi–CCA provides a constant‑memory alternative to replay that consistently outperforms strong baselines , while maintaining zero‑shot and retrieval capablities under domain and prompt shifts. The authors show that geometry drift correlates with performance drops, that their losss is both theoretically justified and efficient, that results are robust to sketch randomness, backbone size, and anchor configurations, and that the overhead and memory footprint remain modest relative to replay methods. While some novelty concerns remain, the way prior methods are applied and validated is in the AC’s view a meaningful contribution.

On balance, AC agrees with positive points raised by all reviewers which outweigh the negative points. The authors are strongly encouraged to include the additional reviewer recommendations, experiments from rebuttal and clarifications in the camera-ready version.

**Reviewer Concerns:**

The rebuttal addresses most technical and empirical concerns in a reasonable way, though some conceptual debates on novelty remain. In particular, the authors add robustness experiments showing low sensitivity to sketch randomness and sketch type, scaling results to ViT‑L backbones and multiple PET configurations, clearer explanations of the CCA certificate, standard deviations and paired t‑tests demonstrating statistical significance of gains, and stronger baselines with heavy regularization and proxy similarity alignment that still underperform Pi–CCA. They also justify the projector loss as a sketched Grassmann distance, clarify that the certificate is updated via stop‑grad EMA+QR for stability, and give additional evidence that prompt invariance and small, label-derived anchor sets are effective and not overly sensitive; they quantify modest overhead and very small certificate storage compared with replay buffers, and provide per‑task results showing consistent gains. These additions target main reviewer concerns and in the AC’s view, resolve them to a satisfactory degree.

**Reviewer Scores:**

Given these clarifications and new experiments, it is likely that Reviewer YmbX and Reviewer TQCn would keep their “6: marginally above the acceptance threshold” scores (YmbX explicitly states they “will keep my score” and TQCn notes that concerns were “gradually being addressed” and asked for t‑tests, which were then provided). Reviewer Z4s6, also at 6, might be further reassured by the overhead and privacy clarifications and per‑task results. Reviewer pV7E, at “4: marginally below the acceptance threshold,” might still view the conceptual novelty as moderate, but the added baselines, geometry–performance correlation, and theoretical justification for the projector loss plausibly reduce the concern that gains are purely due to tuning, and the simplicity of the prompt invariance and anchor construction is better motivated in the constrained replay‑free, constant‑memory setting. Overall, three reviewers lean toward acceptance and one is cautious but not strongly negative, with several of their technical concerns now substantially addressed.

---

### Decision · Program_Chairs · 2026-01-26

Accept (Poster)